# Stimulative Training of Residual Networks: A Social Psychology Perspective of Loafing

**Peng Ye[1][†] , Shengji Tang[1][†] , Baopu Li[2] , Tao Chen[1]*, Wanli Ouyang[3]**
[1]School of Information Science and Technology, Fudan University, [2]Oracle Health and AI, USA,
[3]The University of Sydney, SenseTime Computer Vision Group, Australia, and Shanghai AI Lab
yepeng20@fudan.edu.cn

## Abstract

Residual networks have shown great success and become indispensable in today's deep models. In this work, we aim to re-investigate the training process of residual networks from a novel social psychology perspective of loafing, and further propose a new training strategy to strengthen the performance of residual networks. As residual networks can be viewed as ensembles of relatively shallow networks (i.e., *unraveled view*) in prior works, we also start from such view and consider that the final performance of a residual network is co-determined by a group of sub-networks. Inspired by the social loafing problem of social psychology, we find that residual networks invariably suffer from similar problem, where sub-networks in a residual network are prone to exert less effort when working as part of the group compared to working alone. We define this previously overlooked problem as *network loafing*. As social loafing will ultimately cause the low individual productivity and the reduced overall performance, network loafing will also hinder the performance of a given residual network and its sub-networks. Referring to the solutions of social psychology, we propose *stimulative training*, which randomly samples a residual sub-network and calculates the KL-divergence loss between the sampled sub-network and the given residual network, to act as extra supervision for sub-networks and make the overall goal consistent. Comprehensive empirical results and theoretical analyses verify that stimulative training can well handle the loafing problem, and improve the performance of a residual network by improving the performance of its sub-networks. The code is available at https://github.com/Sunshine-Ye/NIPS22-ST.

## 1 Introduction

Since ResNet [1] wins the first place at the ILSVRC-2015 competition, simple-but-effective residual connections are applied in various deep networks, such as CNN, MLP, and transformer. To explore the secrets behind the success of residual networks, numerous studies have been proposed. He et. al [1] exploit the residual structure to avoid the performance degradation of deep networks. Further, He et. al [2] consider that such a structure can transfer any low level features to high level layers in forward propagation and directly transmit the gradients from deep to shallow layers in backward propagation. Balduzzi et. al [3] find that residual networks can alleviate the shattered gradients problem that gradients resemble white noise. In addition, Veit et. al [4] experimentally verify that residual networks can be seen as a collection of numerous networks of different lengths, namely *unraveled view*. Following this view, Sun et. al [5] further attribute the success of residual networks to shallow sub-networks, which may correspond to the low-degree term when regarding the neural network as a polynomial function. Since the unraveled view is supported both experimentally and

---

*Corresponding Author (eetchen@fudan.edu.cn).   [†]Equal Contribution.

36th Conference on Neural Information Processing Systems (NeurIPS 2022).

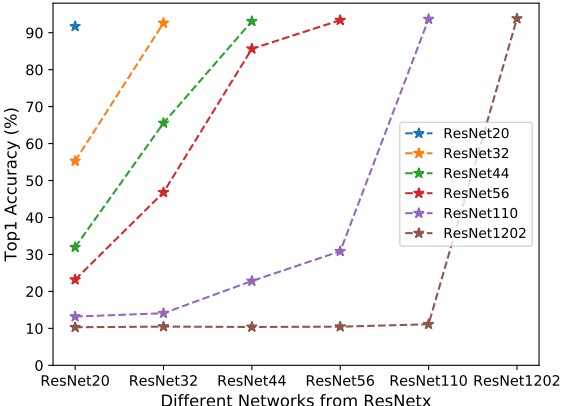

Figure 1: Different residual networks invariably suffer from the problem of network loafing, and deeper residual network tends to have more serious loafing problem. All these networks are trained on CIFAR10 dataset. The horizontal axis means the sampled different sub-networks from ResNetx.

theoretically, we further investigate some interesting mechanism behind residual networks based on this inspiring work. Specifically, we treat a residual network as the ensemble of relatively shallow sub-networks and consider its final performance to be co-determined by a group of sub-networks.

In social psychology, working in a group is always a tricky thing. Compared with performing tasks alone, group members tend to make less efforts when working as part of a group, which is defined as the social loafing problem [6, 7]. Moreover, social psychology researches find that increasing the group size may aggravate social loafing for the decrease of individual visibility [8, 9]. Inspired by these, we find that the ensemble-like networks formed by residual connections also have a similar problem and behavior. As shown in Fig. 1, we can see that, different residual networks invariably suffer from the loafing problem, that is, the sub-networks working in a given residual network are prone to exert degraded performances than these sub-networks working individually. For example, a sub-network ResNet32 in the ResNet56 only has a 46.80% Top1 accuracy, much lower than ResNet32 trained individually with accuracy of 92.63%. Moreover, the loafing problem of deeper residual networks is inclined to be more severe than that of shallower ones, that is, the same sub-network in deeper residual networks constantly presents inferior performance than that in shallower residual networks. For example, ResNet20 within ResNet32 has a 55.28% Top1 accuracy, while ResNet20 within ResNet56 (deeper than ResNet 32) only has a 23.17% Top1 accuracy. Such problems have not been addressed in the literature so far as we know. Hereafter, we define this previously overlooked problem as *network loafing*.[2] As social psychology researches show that social loafing will ultimately cause low productivity of each individual and the collective [10, 11], we consider that network loafing may also hinder the performance of given residual network and all of its sub-networks.

In social psychology, there are two commonly used solutions for preventing social loafing within groups: 1) establishing individual accountability by increasing the *individual supervision* and 2) making tasks cooperative by setting up the overall goal [8, 9]. Inspired by this, we propose a novel training strategy for improving residual networks, namely *stimulative training*. In details, for each mini-batch during stimulative training, besides the main loss of the given residual network in conventional training, we will randomly sample a residual sub-network (individual supervision) and calculate the KL-divergence loss between the sampled sub-network and the given residual network (consistent overall goal). This simple yet effective training strategy can relieve the loafing problem of residual networks, by strengthening the individual supervision of sub-networks and making the goals of residual sub-networks and the given residual network more consistent.

Comprehensive empirical analyses verify that stimulative training can solve the loafing problem of residual networks effectively and efficiently, thus improve both the performance of a given residual network and all of its residual sub-networks by a large margin. Furthermore, we theoretically show the connection of the proposed stimulative training strategy and the improved performance of a given residual network and all of its residual sub-networks. Besides, experiments on various benchmark datasets using various residual networks demonstrate the effectiveness of the proposed training strategy. The contributions of our work can be summarized as the following:

---

[2]*Network loafing* is just a loose analogy to describe a behavior in neural networks that has no strong connection with biology.

- We understand residual networks from a social psychology perspective, and find that different residual networks invariably suffer from the problem of network loafing.
- We improve residual networks from a social psychology perspective, and propose a simple-but-effective stimulative training strategy to improve the performance of the given residual network and all of its sub-networks.
- Comprehensive empirical and theoretical analysis verify that stimulative training can well solve the loafing problem of residual networks.

## 2 Related Works

### 2.1 Unraveled View

As one pioneer work to investigate residual networks, [4] experimentally shows that residual networks can be seen as a collection of numerous networks of different length, namely unraveled view. Subsequently, [5] follows this view and further attributes the success of residual networks to shallow sub-networks first. Besides, [5] considers the neural network as a polynomial function and corresponds shallow sub-network to low-degree term to explain the working mechanism of residual networks. Similar to [4] and [5], this paper also investigates residual networks from the unraveled view. Differently, inspired by social psychology, we further reveal the loafing problem of residual networks under the unraveled view. Besides, we propose a novel stimulative training method to relieve this problem and further unleash the potential of residual networks.

### 2.2 Knowledge Distillation

As a classical method, knowledge distillation [12, 13] transfers the knowledge from a teacher network to a student network via approximating the logits [12, 14, 15] or features [16, 17, 18, 19] output. To avoid the huge cost of training a high performance teacher, some works abandon the naive teacher-student framework, like mutual distillation [20] making group of students learn from each other online, and self distillation [21] transferring knowledge from deep layers to shallow layers. Generally, all these distillation methods need to introduce additional networks or structures, and employ fixed teacher-student pairs. As a comparison, our method does not require any additional network or structure, and the student network is a randomly sampled sub-network of a network. Besides, our method is essentially designed to address the loafing problem of residual networks, which is different from knowledge distillation that aims to obtain a compact network with acceptable accuracy.

### 2.3 One-shot NAS

One-shot NAS is an important branch of neural architecture search (NAS) [22, 23, 24, 25, 26, 27]. Along this direction, [28] trains an once-for-all (OFA) network with progressive shrinking and knowledge distillation to support kinds of architectural settings. Following this work, BigNAS [29] introduces several technologies to train a high-quality single-stage model, whose child models can be directly deployed without extra retraining or post-processing steps. Both OFA and BigNAS aim at simultaneously training and searching various networks with different resolutions, depths, widths and operations. Differently, the proposed method aims at improving a given residual network, thus can be seamlessly applied to the searched model of NAS. As OFA and BigNAS are not designed to solve the loafing problem, their sampling space and supervision signal are also different with the proposed method. More importantly, the social-psychology-inspired problem of network loafing may explain why OFA and BigNAS work.

### 2.4 Stochastic Depth

As a regularization technique, Stochastic Depth [30] randomly disables the convolution layers of residual blocks, to reduce training time and test error substantially. In Stochastic Depth, the reduction in test error is attributed to strengthening gradients of earlier layers and the implicit ensemble of numerous sub-networks of different depths. In fact, its improved performance can be also interpreted as relieving the network loafing problem defined in this work. The theoretical analysis in this work can be also applied to Stochastic Depth. Besides, for better solving the loafing problem, our method samples sub-networks with ordered depth, and uses an additional KL-divergence loss to provide a more achievable target and make the output of a given network and its sub-networks more consistent.

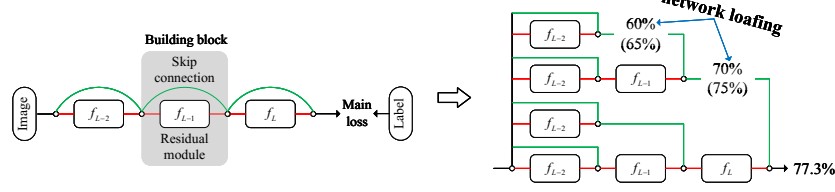

(a) **Common training** scheme suffers from severe **network loafing** problem

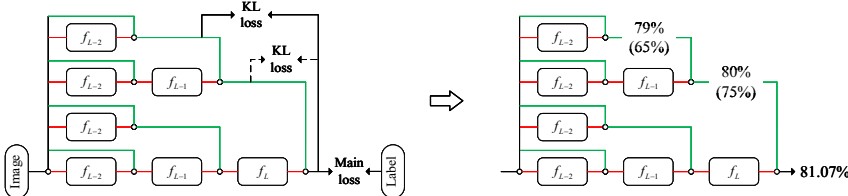

(b) **Stimulative training** scheme can relieve **network loafing** and improve the performance

Figure 2: Illustration of common and stimulative training schemes. Stimulative training can relieve the network loafing problem, and improve the performance of a given residual network (from 77.3% to 81.07%) and all of its sub-networks (e.g., from 60% to 79%). 65% and 75% denote the individual performance of each sub-network.

## 3 Stimulative Training

### 3.1 Motivation

Social loafing is a social psychology phenomenon that individuals lower their productivity when working in a group. Based on the novel perspective that a residual network behaves like an ensemble network [4], we find various residual networks invariably exhibit loafing-like behaviors as shown in Fig. 1 and Fig. 4, which we define as network loafing. As social loafing is "a kind of social disease" that will harm the individual and collective productivity [10], we consider network loafing may also hinder the performance of a residual network and its sub-networks. To alleviate network loafing, it is intuitive to learn from social psychology. There are two common methods for solving social-loafing problem in sociology, namely, establishing individual accountability (i.e., increasing the individual supervision) and making tasks cooperative (i.e., setting up the overall goal) [8, 9]. In order to increase individual supervision, we sample sub-networks in the whole network and provide extra supervision to train each sub-network sufficiently. For the overall goal, we adopt KL divergence loss to constrain the output of sub-networks not far from that of the whole network, which aims at reducing the performance gap and driving the sampled sub-networks to develop cooperatively.

### 3.2 Training Algorithm

In this subsection, we briefly illustrate the working scheme of the proposed stimulative training strategy, and show its difference from the common training method. As shown in Fig. 2, common training only focuses on optimization of main network, thus suffers from severe network loafing, that is, sub-networks lower their performance when working in an ensemble. For example, as shown in Fig. 2(a) sub-networks within the residual network only have an accuracy of 60% and 70%, much lower than their individual accuracy of 65% and 75%. As a comparison, stimulative training optimizes the main network and meanwhile uses it to provide extra supervision for a sampled sub-network at each training iteration, thus well handles the loafing problem. In the test procedure, our method can adopt the main network or any sub-network as the inference model, thus requiring the same or lower memory and inference cost compared with a given residual network.

Formally, for a given residual network to be optimized, we define the main network as $D_m$ and the sub-network as $D_s$. All the sub-networks share weights with the main network, and make up the sampling space $\Theta = \{D_s | D_s = \pi(D_m)\}$, where $\pi$ is sampling operator, usually random sampling. In the training process, we randomly sample a sub-network at each iteration to ensure the whole sampling space can be fully explored. To make the training more efficient and effective, we define a new sampling space obeying ordered residual sampling rule to be discussed in Section 3.3. Denoting $\theta_{Dm}$ and $\theta_{Ds}$ as weights of the main network and the sampled network respectively, $x$ as the mini-batch training sample and $y$ as its label, $\mathcal{Z}$ as the output of network, the total loss of stimulative

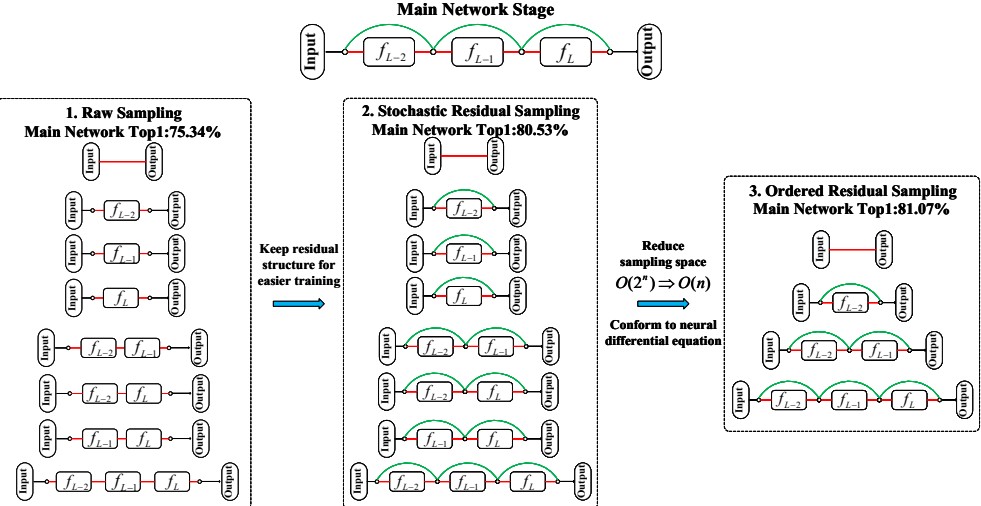

Figure 3: Illustration of ordered residual unraveling rule. Compared with two other sampling strategies, ordered residual sampling can facilitate training and reduce the sampling space to get better main network performance.

training is computed as

$$\mathcal{L}_{mt} = \underbrace{CE(\mathcal{Z}(\theta_{D_m}, x), y)}_{\text{main supervision}} + \underbrace{\lambda KL(\mathcal{Z}(\theta_{D_m}, x), \mathcal{Z}(\theta_{D_s}, x))}_{\text{extra common direction supervision}}, D_s \in \Theta \tag{1}$$

where the $CE$ and $KL$ are the standard cross entropy loss and KL divergence loss. $\lambda$ is the balancing coefficient, and $CE$ loss denotes the main supervision for the main network. $KL$ loss provides an extra supervision for the sampled sub-network, and guarantees all sub-networks to be optimized in the same direction with the main network, which can be considered as setting a common goal for the "ensemble network". We use the standard stochastic gradient descent for updating the model, denoted as $\theta_{D_m}^{t+1} = \theta_{D_m}^t - \eta \frac{\partial \mathcal{L}_{ut}}{\partial \theta_{D_m}}$ , where $\eta$ is the learning rate. Thanks to the weight-sharing between sub-networks and the main network, we only need to update $\theta_{D_m}$ once for each iteration to optimize both the $\theta_{D_m}$ and $\theta_{D_s}$. The pseudo code is shown in Appendix C.1.

### 3.3 Ordered Residual Sampling

Network loafing is based on the novel view that residual networks can be seen as the ensemble of numerous different sub-networks[4]. It is intuitive to adopt raw unraveled view to design sampling space, namely raw sampling. However, there exist two problems in raw sampling. First, it introduces a mass of single branch sub-networks which are hard to optimize, as shown in the left sub-figure of Fig. 3. Second, with the growth of network depth, the size of sampling space increases exponentially. For a residual stage of $n$ blocks, there are $2^n$ sub-networks, which constitutes a too large space to explore. Under limited computation resources, it is difficult to provide sufficient supervision to each sub-network, causing insufficient stimulative training.

Therefore, in order to facilitate stimulative training, we re-design two different sampling ways: stochastic residual sampling and ordered residual sampling. Fig. 3 shows a three-block residual network as a simple illustration for three sampling ways. For raw sampling, we obey the raw definition of [4], that means each sub-network is composed of stacked convolution layers. For stochastic residual sampling, we keep the basic structure of residual blocks and skip some blocks randomly. For ordered residual sampling, we also keep the residual structure but skip blocks orderly, (i.e., always skip the last several blocks). We implement these sampling methods for MobileNetV3 [31] on CIFAR-100. Experimental results show that ordered residual sampling performs better than other two methods (by 5.77% and 0.58% respectively). To explain the superiority of ordered residual sampling, we should pay attention to keeping residual structure and ordered sampling. For keeping residual structure, it can drive the network training process easier and improve the final performance [1]. For ordered sampling, it can noticeably reduce the size of sampling space (from $O(2^n)$ to $O(n)$), making it possible to train each sub-network sufficiently. Besides, all deep networks with residual connections can be approximated by neural differential equations [32], which treat discrete network layers as

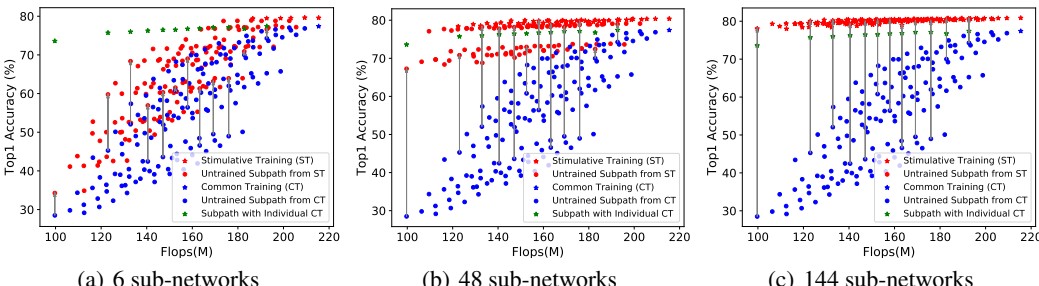

| (a) 6 sub-networks | (b) 48 sub-networks | (c) 144 sub-networks |

Figure 4: Stimulative training using 6, 48, and 144 sub-networks. With more sub-networks involved, the network loafing problem could be relieved better, and the performance of the residual network and all of its sub-networks could be improved more significantly. We train MobileNetV3 with different schemes on CIFAR100 dataset.

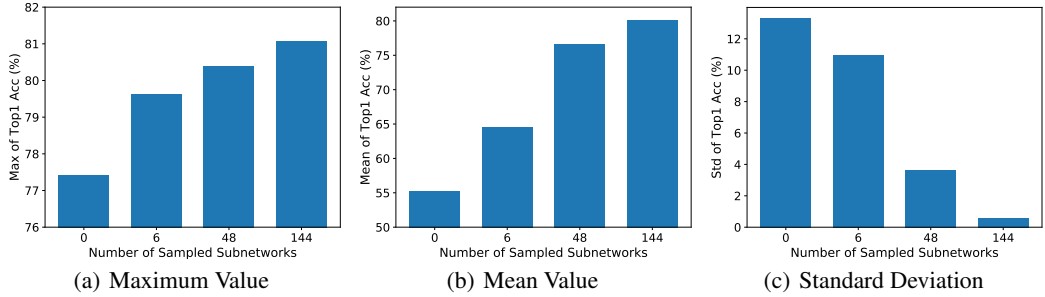

| (a) Maximum Value | (b) Mean Value | (c) Standard Deviation |

Figure 5: (a) Maximum, (b) mean value and (c) standard deviation when training MobileNetV3 with different strategies on CIFAR100 dataset. Sampling 0 sub-networks denotes the common training strategy. Stimulative training with more sub-networks can provide higher maximum, mean and lower standard deviation.

continuous and regard the layer output as the result of adding an integral to the layer input. From this perspective, ordered sampling will maintain the continuity of residual networks. Because of these outstanding features, we will adopt ordered residual sampling in the following experiments.

## 4 Empirical Analysis

In this section, we conduct extensive experiments to analyze and compare the proposed stimulative training and the common training strategy in detail. We first illustrate the loafing problem of residual networks with different training strategies. We further demonstrate statistical characteristics of the performance of the trained residual network and all of its sub-networks. Then, we show the performance drop when destructing trained residual networks at test time, including deleting individual layers, deleting more layers and permuting layers. Besides, we also record the KL distance between a given residual network and all of its sub-networks in the whole training process. For the convenience of analysis, we always use a NAS-searched model MobileNetV3 [31] as the residual network and train it on CIFAR100, unless otherwise specified. The detailed experimental settings can be found in Appendix C.2.

### 4.1 Experiment 1: Checking the loafing problem of residual networks with different training strategies

First, we find that the NAS-searched residual network MobileNetV3 still suffers from the loafing problem. As shown in Fig. 4, when applying the common training strategy to MobileNetV3, the sub-networks from MobileNetV3 (blue solid circle) have much lower accuracies than the same sub-networks individually trained (green five-pointed star), and the performance gap increases when the size of the sub-network decreases. Second, we show that the proposed stimulative training strategy can well handle the loafing problem. As shown in Fig. 4(c), when applying the stimulative training strategy to MobileNetV3, the sub-networks from MobileNetV3 (red five-pointed star) have even better performance than the same sub-networks individually trained (green five-pointed star). More interestingly, as shown in Fig. 4(a), even when we apply stimulative training with 6 sub-networks (red five-pointed star) to MobileNetV3, the performance of all the sub-networks (red solid circle) can

be also greatly improved compared to that of common training (blue solid circle). Besides, As shown in Fig. 4, with more sub-networks involved in stimulative training, the network loafing problem could be relieved better, and the performance of a given residual network and all of its sub-networks could be improved more significantly. More similar results of different datasets and residual networks can be found in Appendix A.

## 4.2 Experiment 2: Collecting statistical characteristics of the performance of all residual sub-networks

To demonstrate that stimulative training can improve the performance of a given residual network and all of its sub-networks, we collect various statistical characteristics of the performance of all residual sub-networks after the stimulative training and common training. As shown in Fig. 5, MobileNetV3 with common training (i.e., sample 0 sub-networks) has the lowest maximum value, mean value, and the highest standard deviation, which is mainly caused by the network loafing problem. As a contrast, MobileNetV3 with stimulative training (i.e., sample 144 sub-networks) has the highest maximum value, mean value, and the lowest standard deviation. Moreover, we can see that, MobileNetV3 using stimulative training with more sub-networks can usually provide higher maximum value, mean value and lower standard deviation. All these results indicate that stimulative training can improve the performance of all residual sub-networks effectively and efficiently.

## 4.3 Experiment 3: Destructing residual networks at test time

In this experiment, we verify that stimulative training can provide stronger robustness in resisting various network destruction operations including deleting individual layers, deleting more layers and permuting layers than common training.

**Deleting individual layers.** We first show the performance drop when deleting individual layers from trained MobileNetV3 at test time. As shown in Fig. 6(a), when applying common training to MobileNetV3, the performance drop is non-negligible when some layers are deleted (especially for the last several layers of each stage), and the trained MobileNetV3 is sensitive to the index of deleted layer. As a comparison, when applying stimulative training to MobileNetV3, no matter which layer is deleted, the performance drop of stimulative trained model is always lower than that of the common trained model, and the performance drop is always smaller than or equal to 1%. Such results verify that stimulative training can reduce the dependence between sub-networks significantly.

**Deleting more layers.** We then show the performance drop when deleting more layers from trained MobileNetV3 at test time. Note that, deleting more layers may come in a variety of combinations, thus we show the statistical box plot of performance drops for all combinations. As shown in Fig. 6(b), when applying common training to MobileNetV3, the medium value, minimum value and maximum value of performance drop for all combinations keeps continuously rising as the number of deleted layers increases. And the difference in performance drop for different combinations when removing the same number of layers can be very large. For example, when deleting 4 layers of common trained MobileNetV3, the Top1 accuracy drop for different combinations can vary from less than 5% to higher than 35%. Differently, when applying stimulative training, no matter how many layers are deleted, the values of performance drop for all combinations always keep very low (i.e., no more than 3%), indicating that stimulative training can bring strong robustness in resisting network destruction.

**Permuting layers.** We further show the performance drop when permuting layers in trained MobileNetV3 at test time. Note that, only the layers in the same stage can be permuted since the resolution and width of different stages are varying, and we show the detailed experimental settings in Appendix C.3. As shown in Fig. 6(c), when applying common training to MobileNetV3, the medium value, minimum value and maximum value of performance drop for all combinations increases with the rising number of permuted layers, and the variance of performance drop for different combinations when permuting the same number of layers may be also very large. As a comparison, the values of performance drop for all permuting combinations always keep very low (i.e., less than 1%), when stimulative training is applied. This further verifies the superiority of stimulative training.

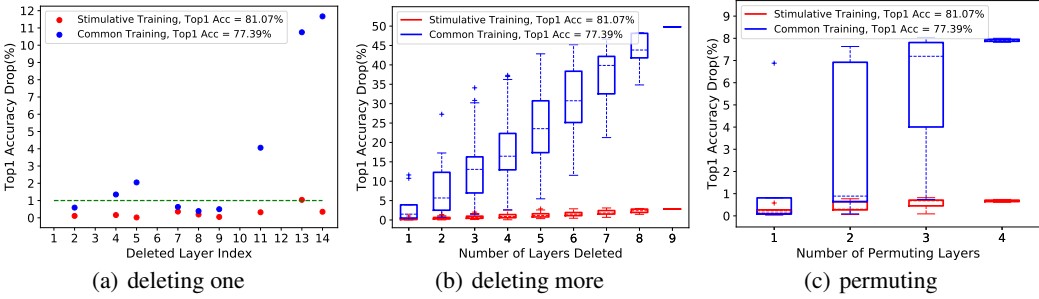

(a) deleting one          (b) deleting more          (c) permuting

Figure 6: Top1 accuracy drop when (a) deleting one layer, (b) deleting more layers and (c) permuting layers from MobileNetV3 with stimulative/common training on CIFAR100 dataset. Stimulative training can provide stronger robustness in resisting various network destruction operations than common training.

### 4.4 Experiment 4: Recording the KL divergence between a residual network and all of its sub-networks when training

To study the consistency between the output of a given residual network and all of its sub-networks, we record their KL distance in the whole training process. As shown in Fig. 7, when applying common training to MobileNetV3, the variance, maximum and medium value of KL distance between a given residual network and all of its sub-networks continuously increases with rising training epoch. For comparison, when stimulative training is applied to MobileNetV3, all values of KL distance between MobileNetV3 and all of its sub-networks keep very small (i.e., no more than 0.25) in the whole training process. Such results imply that the proposed stimulative training can maintain the consistency between the output of a given residual network and all of its sub-networks.

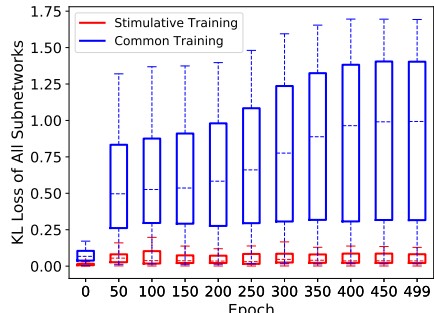

Figure 7: The distribution of KL distance between MobileNetV3 and all of its sub-networks in the whole training process. We train MobileNetV3 on CIFAR100 with stimulative and common schemes.

## 5 Theoretical Analysis

**Analysis 1: Why the performance of sub-networks can be improved.** Based on Eq. 1, our method can be treated as optimizing $CE(\mathcal{Z}(\theta_{D_m}, x), y)$ and $KL(\mathcal{Z}(\theta_{D_m}, x), \mathcal{Z}(\theta_{D_s}, x))$ at the same time. According to the convergence of stochastic gradient descent, after enough number of iterations, both losses can be bounded by a constant, which is formulated as

$$CE(\mathcal{Z}(\theta_{D_m}, x), y) = -\sum_{i=1}^{N} y_i \log p_i^m < \epsilon_1 \tag{2}$$

$$KL(\mathcal{Z}(\theta_{D_m}, x), \mathcal{Z}(\theta_{D_s}, x)) = \mathbb{E}_\Theta \left[ \sum_{i=1}^{N} p_i^m \log \frac{p_i^m}{p_i^s} \right] < \epsilon_2 \tag{3}$$

where $N$ is the number of categories. The $\epsilon_1$ and $\epsilon_2$ are the tiny constants for corresponding losses. $\Theta$ is the set of all sub-networks. The $p^m$ and $p^s$ are the classification probability of main network and sub-network, respectively. Further, we can prove that *the absolute difference of cross entropy of sub-networks and main network can be bounded by a constant* (detailed proof can be found in Appendix B), which can be formulated as

$$|CE(\mathcal{Z}(\theta_{D_m}, x), y) - \mathbb{E}_\Theta \left[ CE(\mathcal{Z}(\theta_{D_s}, x), y) \right]| < \frac{\epsilon_2 + \log N}{e^{-\epsilon_1}} + \epsilon_1 \tag{4}$$

where $\frac{\epsilon_2 + \log N}{e^{-\epsilon_1}} + \epsilon_1$ is a small constant after optimization. Eq. 4 demonstrates that the performance gap between sub-networks and the main network is constrained. Thus, *as the rising of the main network performance, sub-networks performance will be improved step by step.*

Table 1: The main network and all sub-networks performance (%) of stimulative training (ST) compared with common training (CT), on various residual networks and datasets.

| method | MBV3_C10 | MBV3_C100 | Res50_C100 | Res50_Img | Res101_Img |
|---|---|---|---|---|---|
| CT(main) | 95.72 | 77.39 | 76.53 | 76.1 | 77.37 |
| ST(main) | **96.88(+1.16)** | **81.07(+3.68)** | **81.06(+4.43)** | **77.22(+1.12)** | **78.58(+1.21)** |
| CT(all) | 78.13±14.29 | 55.26±13.37 | 34.03±18.92 | 39.46±16.82 | 35.78±18.84 |
| ST(all) | **96.21±0.43** | **80.01±0.59** | **79.66±1.77** | **73.3±2.89** | **75.52±3.6** |

**Analysis 2: Why the performance of given residual networks can be improved.** Neural network training is in fact a form of polynomial regression, and any neural network with activation functions can roughly correspond to a fitted polynomial [33]. For simplifying notations, suppose that both input $x$ and output $y$ in a neural network have only one element. Then, we can consider the training of a neural network as the fitting of a polynomial function.

$$y = F(x) = c_0 + c_1 x + c_2 x^2 + \cdots + c_n x^n + \ldots \tag{5}$$

where $c_1$, $c_2$, $\ldots$, $c_n$, $\ldots$ represent the polynomial coefficients. Based on training samples, we can establish the polynomial (neural network), which denotes the mapping relationship between inputs and outputs. For a test sample $x$, there always exists a point $x_0$ (e.g., a training sample) close enough to $x$, which can obtain an accurate output via the trained model. According to Taylor expansion, $F(x)$ at $x_0$ can be computed as :

$$y = F(x) = \frac{F(x_0)}{0!} + \frac{F'(x_0)}{1!}(x - x_0) + \\ \frac{F''(x_0)}{2!}(x - x_0)^2 + \cdots + \frac{F^{(n)}(x_0)}{n!}(x - x_0)^n + R_n(x) \tag{6}$$

where $R_n(x)$ denotes the higher degree infinitesimal of $(x - x_0)^n$. According to Eq. 5 and Eq. 6, it is obvious that different terms of the fitted polynomial have different effects, and the low-degree terms tend to have greater effects on the predicted performance. Based on this point, [5] further figures out that shallow sub-networks in residual networks roughly correspond to low-degree polynomials consisting of low-degree terms, while deep sub-networks are opposite. Following this theory, the proposed stimulative training can not only *strengthen the learning of different items* of the polynomial (neural network), but also *pay more attention to the learning of low-degree items*. As a result, the final performance of given residual networks can be improved by stimulative training.

## 6 Verification on Various Datasets and Networks

**CIFAR implementation details.** CIFAR[34] is a classical image classification dataset consisting of 50,000 training images and 10,000 testing images. It includes CIFAR-100 in 100 categories and CIFAR-10 in 10 categories. For MobileNetV3 and ResNet50, the data augmentations follow [35], we use SGD optimizer and train the model for 500 epochs with a batch size of 64. The initial learning rate is 0.05 with cosine decay schedule. The weight decay is $3 \times 10^{-5}$ and momentum is 0.9.

**ImageNet implementation details.** We implement our method on large-scale ImageNet[36] dataset containing 1.2 million training images and 50,000 validation images from 1,000 categories. For ResNet families, we use standard data augmentations for ImageNet, as done in [37, 38, 39]. We utilize SGD optimizer to train the model for 100 epochs with a batch size of 512, and the learning rate is 0.2 with cosine decay schedule. The weight decay is $1 \times 10^{-4}$ and momentum is 0.9.

**Main network and sub-networks performance.** We further verify the effectiveness of stimulative training (ST) on various networks and datasets. Table.1 shows its comparison with common training (CT). Besides the Top1 accuracy of main network, we also report the the mean value and standard deviation of Top1 accuracy for all sub-networks. It is obvious that stimulative training can noticeably improve the performance of various networks on various datasets compared to common training (e.g., by 4.43% for ResNet50 on CIFAR-100 and 1.21% for ResNet101 on ImageNet). Moreover, the mean performance of all sub-networks is dramatically improved and standard deviation greatly drops. For example, stimulative training for MobileNetV3 on CIFAR100 has a mean performance of 80.01% (much higher than common training of 55.26%) and a standard deviation of 0.59% (much lower than 13.37%). Such results validate that stimulative training can well alleviate the network loafing problem, and improve the performance of the given residual network and all of its sub-networks. Experimental results on various models and datasets verify the generalization of stimulative training, we advocate utilizing it as a general technology to train residual networks.

## 7   Conclusions

The paper understands and improves residual networks from a social psychology perspective of loafing. Based on the novel view that a residual network behaves like an ensemble network, we find that various residual networks invariably exhibit loafing-like behaviors that are consistent with the social loafing problem of social psychology. We define this previously overlooked problem as network loafing. As the loafing problem hinders the productivity of each individual and the whole collective, we learn from social psychology and propose a stimulative training strategy to solve the network loafing problem. Comprehensive empirical analyses show that stimulative training can improve the performance of a given residual network and all of its sub-networks, and provide strong robustness in resisting various network destruction operations. Furthermore, we theoretically show why the performance of a given residual network and its sub-networks can be improved. Experiments on various datasets and residual networks demonstrate the effectiveness of the proposed method.

## 8   Limitations

The proposed method suffers from about 1.4 times of computation cost of the original model training to get better performance and robustness. As the first research of network loafing problem, the proposed method is a positive pioneer-like exploration. We believe designing a more efficient method to solve the network loafing problem is a worthy research direction in the future.

Residual structure is widely applied in numerous different types of models including DenseNet and transformer. It will be of vital value to study whether the loafing problem exists in these models and explore the proper method to solve this problem. Since most of existing works overlook the loafing problem, it may also be a feasible way to apply the proposed method in pretrained models. What's more, the proposed method adopts main network logits as supervision to alleviate loafing and analogously, the proposed method may be applied in self-supervised learning.

We believe that taking full advantage of splendid achievements in interdisciplinary research can help promote the development of deep learning. We hope that this paper can provide a new perspective to inspire more researchers to comprehend and improve deep neural networks from other fields such as social psychology.

## 9   Acknowledgements

This work is supported by National Natural Science Foundation of China (No. U1909207 and 62071127), and Shanghai Municipal Science and Technology Major Project (No.2021SHZDZX0103). Wanli Ouyang is supported by the Australian Research Council Grant DP200103223, Australian Medical Research Future Fund MRFAI000085, CRC-P Smart Material Recovery Facility (SMRF) – Curby Soft Plastics, and CRC-P ARIA - bionic visual-spatial prosthesis for the blind.

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
