# Stimulative Training of Residual Networks: A Social Psychology Perspective of Loafing

**Peng Ye[1][†] , Shengji Tang[1][†] , Baopu Li[2] , Tao Chen[1]\*, Wanli Ouyang[3]**
[1]School of Information Science and Technology, Fudan University, [2]Oracle Health and AI, USA,
[3]The University of Sydney, SenseTime Computer Vision Group, Australia, and Shanghai AI Lab

## Appendix A: The loafing problem of different datasets and residual networks

We further verify that stimulative training can well handle the loafing problem on different datasets and residual networks. As shown in Fig. r1, we can see that stimulative training can always improve the performance of a given residual network and all of its sub-networks by a larger margin on various residual networks and benchmark datasets. In other words, different residual networks trained on different datasets invariably suffer from the problem of network loafing, which can be well solved by the proposed stimulative training strategy.

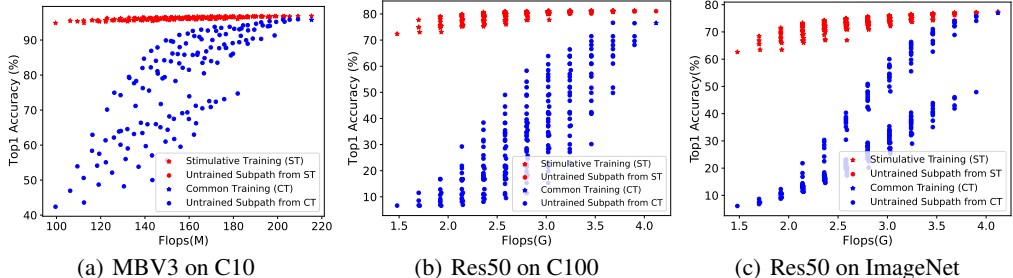

| (a) MBV3 on C10 | (b) Res50 on C100 | (c) Res50 on ImageNet |

Figure r1: Stimulative training can improve the performance of a given residual network and all of its sub-networks significantly. We further verify it on various residual networks and benchmark datasets.

## Appendix B: Proof of theoretical analysis 1

According to the convergence of SGD, cross entropy loss $CE(\mathcal{Z}(\theta_{D_m}, x), y)$ and KL-divergence loss $KL(\mathcal{Z}(\theta_{D_m}, x), \mathcal{Z}(\theta_{D_s}, x))$ used in our method can be bounded by a tiny constant, expressed as

$$CE(\mathcal{Z}(\theta_{D_m}, x), y) = -\sum_{i=1}^{N} y_i \log p_i^m < \epsilon_1, \tag{1}$$

$$KL(\mathcal{Z}(\theta_{D_m}, x), \mathcal{Z}(\theta_{D_s}, x)) = \mathbb{E}_\Theta \left[ \sum_{i=1}^{N} p_i^m \log \frac{p_i^m}{p_i^s} \right] < \epsilon_2, \tag{2}$$

$$0 < p_i^m, p_i^s < 1, \sum_{i=1}^{N} p_i^m = 1, \sum_{i=1}^{N} p_i^s = 1, \tag{3}$$

where $\theta_{D_m}$ and $\theta_{D_s}$ denote the learned weights of the main network and sub-network, respectively. $\mathcal{Z}$ is the network output, $x$ is the input image, and $y$ is the label. $p_i^m$ and $p_i^s$ are the prediction probability of the main network and sub-network, respectively. We define $k$ as the ground truth index (i.e., $y_k = 1$ and $y_i = 0 (i \neq k)$), $\Theta$ as the set of sampled sub-networks, $\epsilon_1$ and $\epsilon_2$ as the tiny constants. Therefore,

---

\*Corresponding Author (eetchen@fudan.edu.cn).   [†]Equal Contribution.

36th Conference on Neural Information Processing Systems (NeurIPS 2022).

our target is to prove that the gap between the $CE$ loss of the main network and all sub-networks is bounded by a tiny constant, written as

$$|CE(\mathcal{Z}(\theta_{D_m}, x), y) - \mathbb{E}_\Theta \left[ CE(\mathcal{Z}(\theta_{D_s}, x), y) \right]| \tag{4}$$

$$= \left| \mathbb{E}_\Theta \left[ \sum_{i=1}^{N} y_i \log \frac{p_i^m}{p_i^s} \right] \right| = \left| \mathbb{E}_\Theta \left[ \log \frac{p_k^m}{p_k^s} \right] \right| < \epsilon_3. \tag{5}$$

Now, we begin to prove the existence of $\epsilon_3$. For Formulation. 1, it can be simplified to $-\log p_k^m \leq \epsilon_1$, which can be equally written as

$$p_k^m \geq e^{-\epsilon_1}. \tag{6}$$

For Formulation. 2, it can be equally written as

$$- H(p^m) - \mathbb{E}_\Theta \left[ \sum_{i=1}^{N} p_i^m \log p_i^s \right] < \epsilon_2, \tag{7}$$

$$- \mathbb{E}_\Theta \left[ \sum_{i=1}^{N} p_i^m \log p_i^s \right] < \epsilon_2 + H(p^m) < \epsilon_2 + \log N, \tag{8}$$

where $H(\cdot)$ represents the entropy operator. Further, based on Formulation. 6 and Formulation. 8, Formulation. 5 can be bounded by

$$\left| \mathbb{E}_\Theta \left[ \log \frac{p_k^m}{p_k^s} \right] \right| = |\log p_k^m - \mathbb{E}_\Theta \left[ \log p_k^s \right]| < |\log p_k^m| + |E_\Theta \left[ \log p_k^s \right]| \tag{9}$$

$$< \epsilon_1 + \left| \frac{E_\Theta \left[ p_k^m \log p_k^s \right]}{p_k^m} \right| < \epsilon_1 + \left| \frac{E_\Theta \left[ \sum_{i=1}^{N} p_i^m \log p_i^s \right]}{p_k^m} \right| \tag{10}$$

$$< \epsilon_1 + \left| \frac{\epsilon_2 + \log N}{p_k^m} \right| < \epsilon_1 + \frac{\epsilon_2 + \log N}{e^{-\epsilon_1}}. \tag{11}$$

In Formulation. 9, we utilize triangle inequality. In Formulation. 10 and 11, we scale the inequality according to Formulation. 8 and 6. As $\epsilon_3 = \epsilon_1 + \frac{\epsilon_2 + \log N}{e^{-\epsilon_1}}$ is a tiny constant which is independent of $p$, we finish the proof.

## Appendix C: More details about experimental settings

### Appendix C.1: The procedure of stimulative training

We show the procedure of stimulative training in Alg. 1. In short, we randomly sample an ordered residual sub-network in the main network in each minibatch, and adopt KL divergence loss to constrain the output of the sub-network not far from that of the main network. Similar to solutions for preventing social loafing in social psychology, sampling ordered residual sub-networks aims to increase individual supervision sufficiently, and adopting KL divergence loss aims to make the goals of residual sub-networks and the given residual network more consistent. We will release our codes upon acceptance of this paper.

---
**Algorithm 1** Stimulative Training

---
**Require:**
    Main network $D_m$; Total training iterations $N$; Loss balanced coefficient $\lambda$;
    Input $x$ and label $y$ of each minibatch; Random sampling $\pi$.
1:  Construct the main network and initialize the main network weights $\theta_{D_m}$.
2:  For each $t \in [1, N]$ do
3:     Sample an ordered residual sub-network $D_s = \pi(D_m)$
4:     Main network forwards $\mathcal{Z}_m = D_m(x, \theta_{D_m})$, and compute the loss
        $\mathcal{L}_m = CE(\mathcal{Z}_m, y)$
5:     Sub-network forwards $\mathcal{Z}_s = D_s(x, \theta_{D_s})$, and compute the loss
        $\mathcal{L}_s = KL(\mathcal{Z}_m, \mathcal{Z}_s)$
6:     Compute the stimulative training loss, $\mathcal{L}_{st} = \mathcal{L}_m + \lambda \mathcal{L}_s$
7:     Backward and update network weights $\theta_{D_m}$ by descending $\nabla_{\theta_{D_m}} \mathcal{L}_{st}$
8:  End.

---

**Appendix C.2: Settings of empirical analysis**

For empirical analysis, we train NAS-searched model MobileNetV3 [1] on CIFAR100 with common and stimulative training strategy, respectively, and test the diverse empirical characteristics of trained MobileNetV3 and all of its ordered residual sub-networks. For training, we use SGD optimizer and train the model for 500 epochs with a batch size of 64, the initial learning rate is 0.05 with cosine decay schedule, the weight decay is $3 \times 10^{-5}$ and momentum is 0.9. For testing, we always employ batchnorm re-calibration for each sampled sub-network following [2].

**Appendix C.3: Settings of destructing residual network**

For the destruction experiment of residual networks, we employ the main body of MobileNetV3 for testing and ignore basic input/output layers, as shown in Table r1. For deleting one or more layers, we may delete every layer except the downsampling layer considering that downsampling layers in MobileNetV3 are single branch structure. For permuting layers, we only permute the layers in the same stage since the resolution and width of different stages in MobileNetV3 are varying. We show the performance drop when conducting different number of permuting operations, and consider all combinations when conducting a fixed number of permuting operations. All possible combinations when conducting different number of permuting operations are shown in Table r2.

Table r1: The main body of MobileNetV3. Input denotes the input size of feature maps. #out is the output channel number and s is the stride. Index denotes the layer index when applying destruction.

| Input | 112²×16 | 56²×24 | 56²×24 | 28²×40 | 28²×40 | 28²×40 | 14²×80 | 14²×80 | 14²×80 | 14²×80 | 14²×112 | 14²×112 | 7²×160 | 7²×160 |
|---|---|---|---|---|---|---|---|---|---|---|---|---|---|---|
| #out | 24 | 24 | 40 | 40 | 40 | 80 | 80 | 80 | 80 | 112 | 112 | 160 | 160 | 160 |
| s | 2 | 1 | 2 | 1 | 1 | 2 | 1 | 1 | 1 | 1 | 1 | 2 | 1 | 1 |
| **Index** | **1** | **2** | **3** | **4** | **5** | **6** | **7** | **8** | **9** | **10** | **11** | **12** | **13** | **14** |

Table r2: All combinations when conducting different number of permuting operations.

| Number of permuting layers | All combinations when permuting layers |
|---|---|
| Number of permuting = 1 | [[[1, 2], [3, 5, 4], [6, 7, 8, 9], [10, 11], [12, 13, 14]],
[[1, 2], [3, 4, 5], [6, 8, 7, 9], [10, 11], [12, 13, 14]],
[[1, 2], [3, 4, 5], [6, 7, 9, 8], [10, 11], [12, 13, 14]],
[[1, 2], [3, 4, 5], [6, 9, 8, 7], [10, 11], [12, 13, 14]],
[[1, 2], [3, 4, 5], [6, 7, 8, 9], [10, 11], [12, 14, 13]]] |
| Number of permuting = 2 | [[[1, 2], [3, 5, 4], [6, 8, 7, 9], [10, 11], [12, 13, 14]],
[[1, 2], [3, 5, 4], [6, 7, 9, 8], [10, 11], [12, 13, 14]],
[[1, 2], [3, 5, 4], [6, 9, 8, 7], [10, 11], [12, 13, 14]],
[[1, 2], [3, 4, 5], [6, 8, 7, 9], [10, 11], [12, 14, 13]],
[[1, 2], [3, 4, 5], [6, 7, 9, 8], [10, 11], [12, 14, 13]],
[[1, 2], [3, 4, 5], [6, 9, 8, 7], [10, 11], [12, 14, 13]],
[[1, 2], [3, 4, 5], [6, 8, 9, 7], [10, 11], [12, 13, 14]],
[[1, 2], [3, 4, 5], [6, 9, 7, 8], [10, 11], [12, 13, 14]],
[[1, 2], [3, 5, 4], [6, 7, 8, 9], [10, 11], [12, 14, 13]]] |
| Number of permuting = 3 | [[[1, 2], [3, 5, 4], [6, 8, 7, 9], [10, 11], [12, 14, 13]],
[[1, 2], [3, 5, 4], [6, 7, 9, 8], [10, 11], [12, 14, 13]],
[[1, 2], [3, 5, 4], [6, 9, 8, 7], [10, 11], [12, 14, 13]],
[[1, 2], [3, 5, 4], [6, 8, 9, 7], [10, 11], [12, 13, 14]],
[[1, 2], [3, 5, 4], [6, 9, 7, 8], [10, 11], [12, 13, 14]],
[[1, 2], [3, 4, 5], [6, 8, 9, 7], [10, 11], [12, 14, 13]],
[[1, 2], [3, 4, 5], [6, 9, 7, 8], [10, 11], [12, 14, 13]]] |
| Number of permuting = 4 | [[[1, 2], [3, 5, 4], [6, 8, 9, 7], [10, 11], [12, 14, 13]],
[[1, 2], [3, 5, 4], [6, 9, 7, 8], [10, 11], [12, 14, 13]]] |

**Appendix C.4: Experimental settings of Fig. 1 in the manuscript**

For ResNetx on CIFAR10 dataset, we directly utilize the pretrained models in a publicly available pytorch repository [3]. This repository provides a valid implementation that matches with the description of the original paper [4], with comparable or smaller test error. When testing, we always employ batchnorm re-calibration for sampled sub-networks following [2].

## Appendix D: The trajectory of training loss and test accuracy

We show the trajectory of training loss and test accuracy when applying stimulative and common training in Fig. r2. The point of the highest Top1 accuracy is represented by five-pointed star for both approaches. As we can see, on different residual networks and benchmark datasets, stimulative training can always yield higher accuracy than common training (i.e., 96.88% vs 95.72%, 81.07% vs 77.39%, 81.06% vs 76.53%). Furthermore, we observe that the curves of Top1 accuracy and training loss of residual networks with common training rapidly approach flat, while stimulative training continuously reduces the training loss and increases the Top1 accuracy.

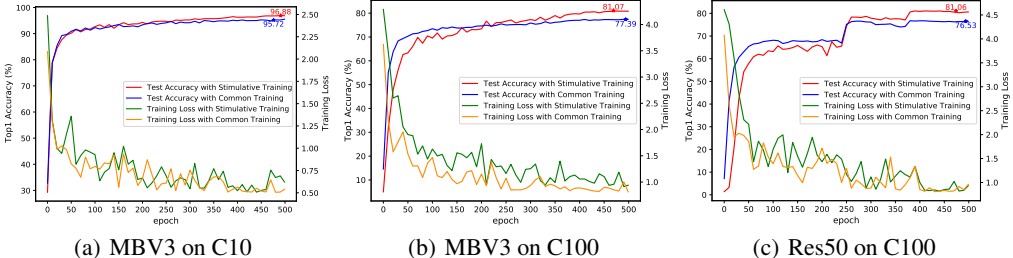

| (a) MBV3 on C10 | (b) MBV3 on C100 | (c) Res50 on C100 |

Figure r2: The trajectory of training loss and test accuracy when applying stimulative and common training on different residual networks and benchmark datasets.

## Appendix E: Ablation study of the balanced coefficient $\lambda$

We show the ablation study of the balanced coefficient $\lambda$ between cross entropy loss and KL divergence loss in Fig. r3. As we can see, on different residual networks and benchmark datasets, stimulation training with different balance coefficients may yield slightly different Top1 accuracies, but all these results are much better than those obtained by common training. In addition, as shown in Fig. r3, the optimal balance coefficients for MobileNetV3 on CIFAR10, MobileNetV3 on CIFAR100 and ResNet50 on CIFAR100 are 5, 10 and 10 respectively. Similar results can be found for ResNet families on ImageNet, and we find that their optimal balance coefficients are 1.

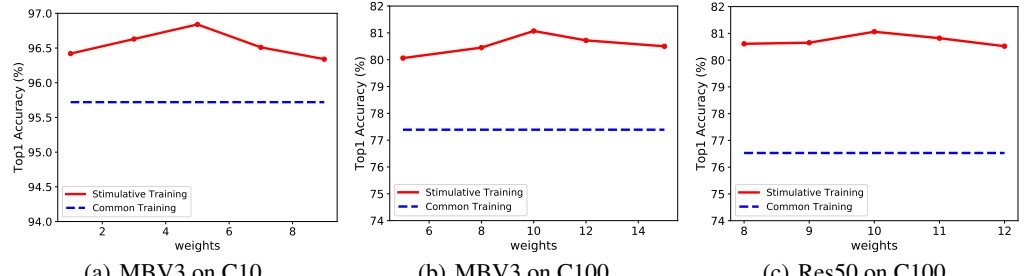

| (a) MBV3 on C10 | (b) MBV3 on C100 | (c) Res50 on C100 |

Figure r3: Ablation study of the balanced coefficient $\lambda$ between cross entropy loss and KL divergence loss on different residual networks and benchmark datasets.

## Appendix F: Rebuttle additional part

### Appendix F.1: Comparison with different methods

We further compare the proposed stimulative training with self distillation [5], stochastic depth [6], and common training with providing supervision directly to each layer or stage. We train MobileNetV3 with these methods on CIFAR100 dataset. For a fair comparison, we adopt the same basic training settings such as training epoch, data augmentation, learning rate and optimizer, which has been shown in the manuscript Section.6. The detailed respective training settings are given as follows.

For self distillation we adopt the similar hyper-parameters as [5] and fine tune them for MobileNetV3. In the logits distillation, the main loss coefficient is 0.8 and distillation coefficient is 0.2. In the feature distillation, the feature distillation coefficient of each stage is 0.02.

For stochastic depth, due to forbidding skipping down sampling layer in MobileNetV3, we ensure the first layer of each stage won't be dropped. The survive rate of layers are declined linearly [6] and the final survive rate p=0.9.

For layer and stage supervision training, we introduce additional transforming heads to project the features of stages or layers to the same dimension space and utilize cross entropy to generate supervision. The stage feature loss coefficient is [0.13, 0.2, 0.27, 0.4] and layer feature loss coefficient is the corresponding stage coefficient divided by the layer number of stage.

The comprehensive comparisons are shown in Table r3.As we can see, layer supervision and stochastic depth can improve both the performance of the main network and the average performance of all subnetworks, stage supervision and self-distillation can only improve the performance of the main network, while the proposed stimulative training can achieve the highest performance of main network and the highest average performance of all subnetworks. Besides, as shown in Fig r8, the proposed stimulative training can better relieve the network loafing problem than all the other methods. As shown in Fig r4, Fig r5, Fig r6 and Fig r7, the proposed stimulative training can provide stronger robustness in resisting various network destruction operations than all the other methods.

Besides above experimental results, we find that: 1) The improved performance of stochastic depth can be also interpreted as relieving the loafing problem defined in this work; 2) the proposed stimulative training is actually complementary to layer/stage supervision and self-distillation, and their combinations can be a worthy research direction in the future.

### Appendix F.2: Loafing of DenseNet networks

To show that the loafing problem are replicable, we further verify that DenseNet also suffers from the loafing problem on ImageNet and CIFAR100. We follow [7] to select 4 typical networks (DenseNet121, 169, 201, 264), in which the tinier one is completely the sub-network of the larger one, to validate the loafing problem. For ImageNet, we just apply the pretrained weights of DenseNet121, 169, 201 and test the sub-network from larger one by succeeding the weights. For CIFAR100, we train all 4 networks utilizing standard training setting and do the same test above. For details, the optimizer is SGD and the initial learning rate is 0.1 and divided by 5 at 60th, 120th, 160th for 200 epochs with batchsize 128, weight decay 5e-4, Nesterov momentum of 0.9.

Table r5 and Table r6 show the results of different DenseNet networks which are trained on ImageNet and CIFAR100 respectively. As we can see that, different DenseNet networks invariably suffer from the loafing problem, that is, the sub-networks working in a given DenseNet network are prone to exert degraded performances than these sub-networks working individually. Moreover, the loafing problem of deeper DenseNet networks is inclined to be more severe than that of shallower ones, that is, the same sub-network in deeper DenseNet networks constantly presents inferior performance than that in shallower DenseNet networks. As Fig 1 of the main text has shown that different ResNet networks invariably suffer from the loafing problem, we can conclude that various residual networks invariably suffer from the loafing problem on various datasets.

### Appendix F.3: Training time and memory cost

We further show the difference in training time and memory consumption between common training (CT) and stimulative training (ST) in Table r4. For training time, ST is about 1.4 times (but less than 2 times) that of CT, since ST employs the main network and a sampled subnetwork at each step and the sampled subnetwork usually takes much less time than the main network. For memory consumption, ST and CT are basically the same, since each subnetwork is sampled from the main network.

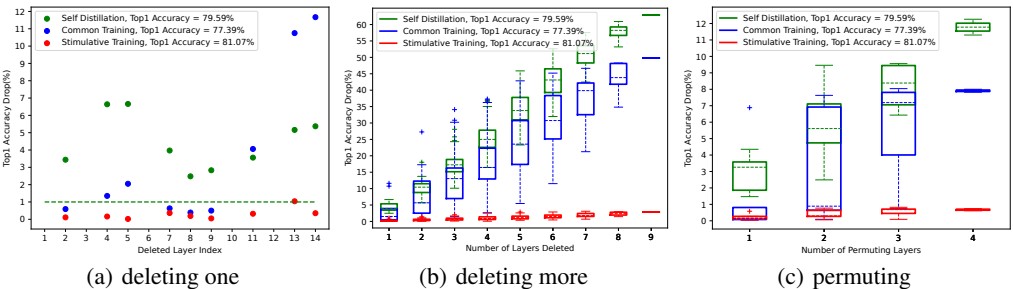

|        (a) deleting one        |        (b) deleting more        |        (c) permuting        |

Figure r4: Top1 accuracy drop when (a) deleting one layer, (b) deleting more layers and (c) permuting layers from MobileNetV3 with stimulative/common/self distillation training on CIFAR100 dataset.

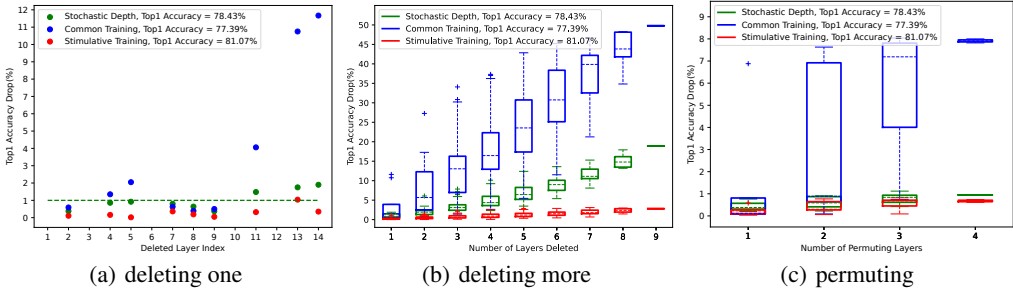

(a) deleting one        (b) deleting more        (c) permuting

Figure r5: Top1 accuracy drop when (a) deleting one layer, (b) deleting more layers and (c) permuting layers from MobileNetV3 with stimulative/common/stochastic depth training on CIFAR100 dataset.

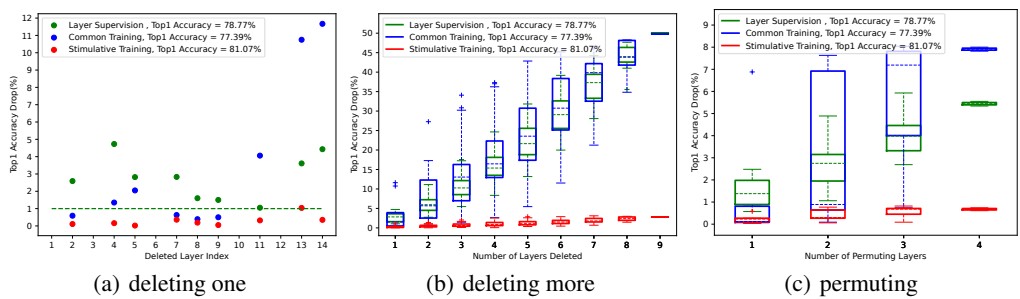

(a) deleting one        (b) deleting more        (c) permuting

Figure r6: Top1 accuracy drop when (a) deleting one layer, (b) deleting more layers and (c) permuting layers from MobileNetV3 with stimulative/common/layer supervision training on CIFAR100 dataset.

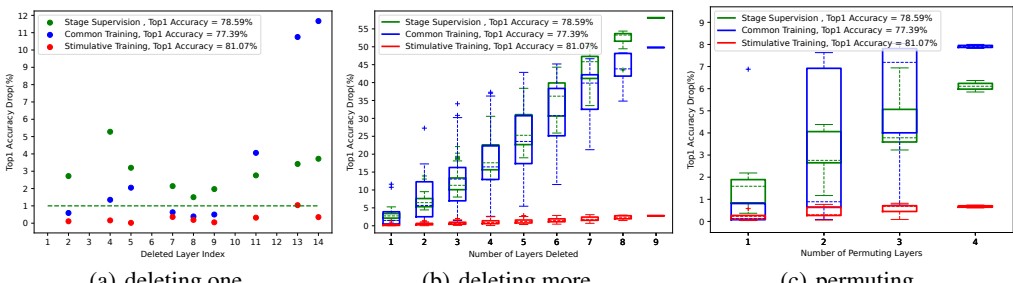

(a) deleting one        (b) deleting more        (c) permuting

Figure r7: Top1 accuracy drop when (a) deleting one layer, (b) deleting more layers and (c) permuting layers from MobileNetV3 with stimulative/common/stage supervision training on CIFAR100 dataset.

Table r3: Comprehensive comparisons among different methods, including common training (CT), stimulative training (ST), common training (CT) with layer/stage supervision, Self-Distillation and Stochastic Depth.

| Method | Time | Memory | Main(%) | All(%) |
|---|---|---|---|---|
| CT | 16.91h | 3291MiB | 77.39 | 55.26±13.37 |
| CT+layer supervision | 23.3h | 7193MiB | 78.77 | 59.18±11.12 |
| CT+stage supervision | 19.3h | 5197MiB | 78.59 | 54.82±13.31 |
| Self distillation | 26.8h | 3887MiB | 79.59 | 50.39±14.22 |
| Stochastic depth | **16.9h** | **3291MiB** | 78.43 | 70.72±3.76 |
| ST | 24.08h | **3291MiB** | **81.07** | **80.01±0.59** |

Table r4: Training time and memory consumption on different models and datasets. CT and ST denote common training and stimulative training, respectively.

| Method | MBV3_C10 | MBV3_C100 | Res50_C100 |
|---|---|---|---|
| CT (time) | 16.77h | 16.91h | 15.28h |
| ST (time) | 23.64h | 24.08h | 21.52h |
| CT (memory) | 3361MiB | 3291MiB | 4647MiB |
| ST (memory) | 3361MiB | 3291MiB | 4647MiB |

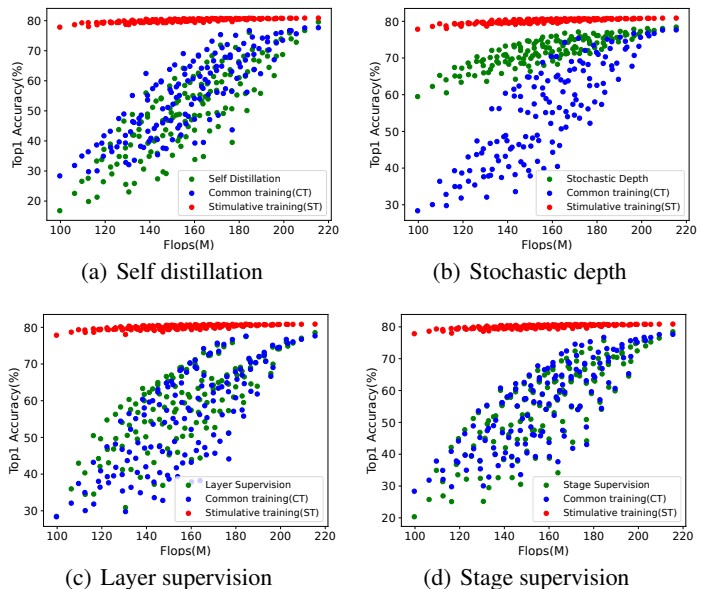

(a) Self distillation  (b) Stochastic depth

(c) Layer supervision  (d) Stage supervision

Figure r8: Comparisons of sub-networks performance on MBV3 with different training methods. As we can see, the proposed stimulative training can better relieve the network loafing problem than all the other methods.

Table r5: Different DenseNet networks trained on ImageNet invariably suffer from the problem of network loafing.

| Main-net\Sub-net | DenseNet121 | DenseNet169 | DenseNet201 |
|---|---|---|---|
| DenseNet121 | 74.86 | 20.91 | 11.57 |
| DenseNet169 | - | 76.46 | 51.18 |
| DenseNet201 | - | - | 77.44 |

Table r6: Different DenseNet networks trained on CIFAR100 invariably suffer from the problem of network loafing.

| Main-net\Sub-net | DenseNet121 | DenseNet169 | DenseNet201 | DenseNet264 |
|---|---|---|---|---|
| DenseNet121 | 78.84 | 43.64 | 31.51 | 10.01 |
| DenseNet169 | - | 79.64 | 70.78 | 48.41 |
| DenseNet201 | - | - | 79.77 | 62.29 |
| DenseNet264 | - | - | - | 79.81 |