# OpenReview forum: "Stimulative Training of Residual Networks: A Social Psychology Perspective of Loafing"
_NeurIPS.cc/2022/Conference — NeurIPS 2022 Accept_

### Official Review · Reviewer_gf27 · 2022-06-30

**Rating:** 4
**Confidence:** 5
**Soundness:** 3 good
**Presentation:** 3 good
**Contribution:** 3 good

**Summary:**

The authors argue that subnetworks in ResNet tend to perform sub-optimally due to a "social loafing effect". That is, since no single subnetwork is accountable for the final performance, they all tend to rely on the aggregate group performance rather than achieving competitive individual performance. In order to alleviate this problem, the authors propose including an additional loss term that encourages subnetworks to match the predictions of the group.  In experiments the authors show that not only their loss solves the loafing problem but also improves the overall final performance. An additional theoretical analysis shows that improving subnetworks is tied to improving overall network performance.

**Questions:**

* Does loafing happen in stochastic depth networks? In fact, it would be nice to stochastic depth networks as a baseline in all the figures.
* Can your method be applied to pre-trained neural networks?
* What if instead of the KL you used the same cross-entropy loss as in the output? That might reduce the computational cost since you would be able to compute the subnetwork loss without the logits of the main network.
* It would be interesting if you checked whether DenseNet also suffers from loafing.
* If the block dropping strategy is the same as in stochastic depth networks, you could just cite them and use the space to introduce changes suggested by the reviewers.
* A main limitation is the additional computational cost, could you comment on it? Another option is to find a better strategy that does not increase it (e.g. keeping a buffer of previous logits so that you do not have to execute the main network every time).

**Minor**
* L48: The loafing problem.. inclines to be more severe -> is inclined to be
* L142: sub-networks.. only has -> only have
* L157: balanced coefficient -> balancing coefficient
* L161: To open and close quotes (") in latex use `` blabla '', this way they display properly in the text
* L34: unraveled view -> the unraveled view

**Limitations:**

There is no limitations section, I encourage the authors to add one discussing issues such as the computational cost or the applicability of their method beyond resnets (such as transformers).

**Strengths And Weaknesses:**

Overall Review
===========
Overall, the authors present an interesting phenomenon in residual networks in an ingenious way (social loafing), while making use of that knowledge to improve such architecture. On the other hand, the experimental evaluation leaves many questions open (see below). In its current state, I think this work does not meet NeurIPS standards but I encourage the authors to continue working on it and to submit a rebuttal.

Strengths
=======
* The authors succeed at showing that "social loafing" does happen in residual networks and that fixing it results in a performance improvement.
* The experimental design succeeds at convincing the reader that there's loafing and that it can be fixed.
* The authors include a theoretical analysis and reproducibility details in the Appendix.

Weaknesses
=========
* Reading the submission generates a series of questions that are not answered by this work. For instance, does loafing happen in stochastic depth networks? In fact, it would be nice to stochastic depth networks as a baseline in all the figures. Can your method be applied to pre-trained neural networks? What if instead of the KL you used the same cross-entropy loss as in the output?
* The authors of DenseNet [A] state that this kind of neural network is less redundant than resnet since each layer is aware of what the other layers are learning. It would be interesting if you checked whether DenseNet also suffers from loafing.
* The block dropping strategy is the same as in stochastic depth networks, thus I would devote less space to it in the paper and state that you use the same strategy.
* While stochastic depth networks reduce the training computation time, this work increases it.

[A] Huang, Gao, et al. "Densely connected convolutional networks." Proceedings of the IEEE conference on computer vision and pattern recognition. 2017.

Detailed comments
===============
Originality
------------
I found original the connection with social loafing. The method to solve it is a combination of multiple existing ideas but I think that it can be considered novel given the context in which it is introduced.

Quality
--------
The overall quality is good but the experimental quality could be improved. For example, you could explore if loafing happens or if it is solved with other methods such as stochastic depth or densenet.

Clarity
-------
The text is clear and easy to read. There are some minor typos.

Significance
--------------
Given the performance improvement introduced by the proposed method, it could be adopted by future work. A major drawback is the increase in computational cost due to having to compute two losses every time.

Reproducibility
-----------------
The authors provide details to reproduce their work, however they do not provide the final code.

---

> ### Author Response · Authors · 2022-08-02
> **Response to Reviewer  gf27**
>
> ***Q5:block dropping strategy***
>
> A5: The block dropping strategy in stochastic depth corresponds to Stochastic Residual Sampling in this work. In Fig. 3 of the main text, we show that Ordered Residual Sampling (81.07%) performs better than Stochastic Residual Sampling (80.53%) for stimulative training. The main reason is that, the former can noticeably reduce the size of sampling space (from O(2^n) to O(n)), making it possible to train each sub-network sufficiently under limited computation resources. In addition, from the perspective of neural differential equations [2], Ordered Residual Sampling will maintain the continuity of residual networks.
>
> [2] Chen R T Q, Rubanova Y, Bettencourt J, et al. Neural ordinary differential equations[J]. Advances in neural information processing systems, 2018, 31.
>
> ***Q6:additional computational cost***
>
> A6: Thanks for the helpful suggestion. Following the reviewer’s suggestion, we further show the training time and memory consumption of stimulative training (ST) and common training (CT) in Table r5. For training time, ST is about 1.4 times that of CT, stochastic depth is about 0.75 times that of CT [3], since the sampled subnetwork usually takes much less time than the main network. For memory consumption, stochastic depth, ST and CT are basically comparable, since each subnetwork is sampled from the main network. For the performance, compared to stochastic depth, ST can achieve higher performance of main network and higher average performance of all subnetworks (as shown in Table r1), better relieve the network loafing problem (as shown in Fig. r8 (b) of the revised supplementary), and provide stronger robustness in resisting various network destruction operations (as shown in Fig. r5 of the revised supplementary). For a better strategy, keeping a buffer of previous logits (so that we do not have to execute the main network every time) could be a good solution to reduce the computational cost, but it may affect the performance since the goals of sub-networks are not consistent with the main network anymore. Moreover, designing a more efficient method to solve the network loafing problem is a worthy research direction in the future.
>
> [3] Huang G, Sun Y, Liu Z, et al. Deep networks with stochastic depth[C]//European conference on computer vision. Springer, Cham, 2016: 646-661.
>
> **Table r5: Training time and memory consumption**
> |Method |	MBV3_C10|	MBV3_C100|	Res50_C100|
> |:------ |:-----------|:---|:---|
> |CT (time)|	16.77h	|16.91h	|15.28h|
> |ST (time)|	23.64h|	24.08h|	21.52h|
> |CT (memory)|	3361MiB	|3291MiB|	4647MiB|
> |ST (memory)|	3361MiB	|3291MiB|	4647MiB|
>
> ***Q7: Minor issues and code***
>
> A7: Following the reviewer’s suggestion, we have fixed these typos in the revised manuscript. Besides, we have carefully checked the manuscript, avoiding typos and errors to our best. We will release our codes upon acceptance of this paper.
>
> ***Q8: limitations section***
>
> A8: Thanks for the helpful suggestion, we have added the limitations section in the revised supplementary to discuss possible issues, applications and societal impact of the proposed method.
>
> The proposed method suffers from about 1.4 times of computation cost of the original model training to get better performance and robustness. As the first research of network loafing problem, the proposed method is a positive pioneer-like exploration. We believe designing a more efficient method to solve the network loafing problem is a worthy research direction in the future.
>
> Residual structure is widely applied in numerous different types of models including DenseNet and transformer. It will be of vital value to study whether the loafing problem exists in these models and explore the proper method to solve this problem. Since most of existing works overlook the loafing problem, it may also be a feasible way to apply the proposed method in pretrained model. What’s more, the proposed method adopts main network logits as supervision to alleviate loafing and analogously, the proposed method may be applied in self-supervised learning.
>
> We believe taking full advantage of splendid achievements in interdisciplinary research can help promote the development of deep learning. We hope this paper can provide a new perspective to inspire more researchers to comprehend and improve DNNs from other fields such as social psychology.

---

> > ### Comment · Reviewer_gf27 · 2022-08-09
> > **Reply to authors**
> >
> > Thank you for your responses and good job with the rebuttal! After reading the other reviews and responses (particularly to reviewer 6eTg) I think it would be important that you clarify in the text that social loafing is just a metaphor to describe a behavior in neural networks that has no strong connection with biology. Overall I still support the acceptance of your work.

---

> > > ### Author Response · Authors · 2022-08-09
> > > **Authors Response After Rebuttal**
> > >
> > > We appreciate your valuable and insightful comments. We feel glad about your generally favorable assessment of our methodology. We will clarify in the final version that "social loafing is just a metaphor to describe a behavior in neural networks that has no strong connection with biology".

---

> ### Author Response · Authors · 2022-08-02
> **Response to Reviewer gf27**
>
> Thanks for your review comments.
>
> ***Q1:Does loafing happen in stochastic depth networks?***
>
> A1:Following the reviewer’s suggestion, we further check whether the loafing problem happens in Stochastic Depth networks: (1) The comprehensive comparisons are shown in Table r1. As we can see, Stochastic Depth does improve the performance of the main network and the average performance of all subnetworks, but stimulative training can achieve much higher performance of main network and much higher average performance of all subnetworks. (2) As shown in Fig r8 (b) of the revised supplementary, Stochastic Depth does relieve the loafing problem, but stimulative training can better handle this problem. (3) As shown in Fig. r5 of the revised supplementary, Stochastic Depth also provides some robustness in resisting various network destruction operations, but stimulative training can provide stronger robustness.
>
> From these experimental results, we can draw two conclusions. On the one hand, the improved performance of Stochastic Depth can be also interpreted as relieving the loafing problem defined in this work, and the theoretical analysis in the main text can be also applied to Stochastic Depth. On the other hand, the proposed stimulative training can better solve the network loafing problem, by sampling sub-networks with ordered depth, using an additional KL-divergence loss to provide a more achievable target, and making the output of a given network and its sub-networks more consistent.
>
> **Table r1: Comparisons**
> |Method|Main(%)|All(%)|
> |:------ |:-----------|:----|
> |CT|	77.39|	55.26±13.37|
> |Stochastic Depth|	78.43|	70.72±3.76|
> |ST|	81.07|	80.01±0.59|
>
> ***Q2:Can your method be applied to pre-trained neural networks?***
>
> A2: Our method is simple and general. Technically, it can be applied to pre-trained neural networks. Since the rebuttal time is very tight and this work focuses on understanding and improving residual networks from a social psychology perspective, we leave the exploration of this topic for future research.
>
> ***Q3:What if instead of the KL you used the same cross-entropy loss as in the output?***
>
> A3: Following the reviewer’s suggestion, we further try experiments with the same cross-entropy loss as the output instead of the KL. Specifically, we randomly sample a subnetwork by ordered residual sampling (ORS) at each step and use the same cross-entropy (CE) loss to update the parameters of each subnetwork. The results are shown in Table r2. Although the computational cost is reduced (since we don’t need the logits of the main network), the performance is also affected and even lower than common training. We consider there are three reasons. First, subnetworks are randomly sampled rather than obtained by linear decay rule [1]. Second, the supervision signal of stimulative training is more achievable for sampled subnetworks and is continuously improved throughout the training process. Third, the goals of sub-networks are not consistent with the main network anymore, but a consistent overall goal is important for solving the loafing problem in social psychology.
>
> [1] Huang G, Sun Y, Liu Z, et al. Deep networks with stochastic depth[C]//European conference on computer vision. Springer, Cham, 2016: 646-661.
>
> **Table r2: Comparisons**
> |Method|MBV3_C100|
> |:------ |:-----------|
> |CT	|77.39|
> |ORS+CE	|76.95|
> |ST	|81.07|
>
> ***Q4:It would be interesting if you checked whether DenseNet also suffers from loafing.***
>
> A4: Following the reviewer’s suggestion, we further check whether DenseNet also suffers from the loafing problem. Table r3 and Table r4 show the results of different DenseNet networks which are trained on ImageNet and CIFAR100 respectively. As we can see that, different DenseNet networks invariably suffer from the loafing problem, that is, the sub-networks working in a given DenseNet network are prone to exert degraded performances than these sub-networks working individually. Moreover, the loafing problem of deeper DenseNet networks is inclined to be more severe than that of shallower ones, that is, the same sub-network in deeper DenseNet networks constantly presents inferior performance than that in shallower DenseNet networks. As Fig 1 of the main text has shown that different ResNet networks invariably suffer from the loafing problem, we can conclude that various residual networks invariably suffer from the loafing problem on various datasets.
>
> **Table r3:DenseNet networks trained on ImageNet**
> |Main-net\Sub-net|	DenseNet121	|DenseNet169|	DenseNet201|
> |:------ |:-----------|:---|:---|
> |DenseNet121|	74.86|	20.91|	11.57|
> |DenseNet169	|-|	76.46|	51.18|
> |DenseNet201	|-	|-|	77.44|
>
> **Table r4:DenseNet networks trained on CIFAR100**
> |Main-net\Sub-net|	DenseNet121	|DenseNet169|	DenseNet201|	DenseNet264|
> |:------ |:-----------|:---|:---|:---|
> |DenseNet121|	78.84|	43.64|	31.51|	10.01|
> |DenseNet169|	-|	79.64|	70.78|	48.41|
> |DenseNet201|	-|	-|	79.77|	62.29|
> |DenseNet264|	-|	-|	-|	79.81|

---

### Official Review · Reviewer_6eTg · 2022-07-11

**Rating:** 3
**Confidence:** 3
**Ethics Flag:** Yes
**Soundness:** 2 fair
**Presentation:** 3 good
**Contribution:** 2 fair

**Summary:**

The paper proposes a view on residual network training from a social psychology perspective. An analogy is made where individual sub-networks of a ResNet are compared to people that are assigned a group project. Social loafing is the phenomenon that work is often not equally distributed in such groups. The paper argues that sub-networks do not perform equally on the classification task in a similar fashion and proposes two mechanisms (random sampling of blocks, and KL divergence between intermediate representations) as a remedy.

**Questions:**

The questions mainly follow the weaknesses above.
1. Is the grounding of the work in social psychology fundamentally important to the presented approach?

2. How can the objective of learning redundant residual blocks be combined with the understanding that CNNs extract feature hierarchies?

3. How does the method differ in performance to previous methods with similar ideas?



**Ethics Review Area:**

["I don’t know"]

**Limitations:**

Limitations and societal impact are not discussed in the paper. Especially because of the framing as inspired by social psychology, it would be very important to discuss both aspects in the paper.


**Strengths And Weaknesses:**

Strengths

The results show that the proposed mechanisms do improve the performance of the overall network and also individual sub-networks.

The writing of the paper is clear and the goal, method and results are presented in an easily understandable format.

The paper contains many results of training networks with error bars which has high computational load.

Weaknesses

Framing: The grounding of the work in social psychology is extremely far fetched. ResBlocks are compared to people in a complex social group setting. The paper itself compares training a network to polynomial regression which has nothing to do with the complex social dynamics in a work group. There are no insights gained from this analogy and comparing a couple of ResBlocks to people contributes to false AI hype and other ethical issues. I strongly advocate for removing this framing from the paper.

Learning redundant blocks: A large portion of the analysis is aimed at the performance of individual sub networks, permuting and deleting residual blocks and the KL divergence between intermediate representations. In all these experiments, the proposed training strategy shows better results than conventional training. This is not very surprising as the additional objective of adding a KL divergence term between representations and sampling sub-networks trains the model exactly for these tasks. Moreover, both objectives essentially train the individual block to be redundant. (This is another breakdown point for the loafing analogy where the optimal group work outcome is not achieved when every member does the same job) It is questionable if this is a desired behavior as CNNs are commonly understood to learn a hierarchical structure of representations. If blocks can be reshuffled or deleted without affecting the performance much, this structure is lost. It is possible that this effect is less important on small images such as CIFAR100.

Comparisons: The method is only compared to conventional training. Similar approaches have been explored that add auxiliary losses, distill representation (student teacher, KD, [12-19]) or drop residual blocks([22]) and it would be important to understand their effect on the network in comparison to the proposed approach. This would also make the comparison of dropping and reshuffling blocks more interesting.

---

> ### Author Response · Authors · 2022-08-02
> **Response to Reviewer 6eTg**
>
> ***Q3: Comparisons. How does the method differ in performance to previous methods with similar ideas?***
>
> A3: Following the reviewer’s suggestion, we further compare the proposed stimulative training with different methods including layer/stage supervision, Self-Distillation [7] and Stochastic Depth [8]: (1) The comprehensive comparisons are shown in Table r1. As we can see, layer supervision and stochastic depth can improve both the performance of the main network and the average performance of all subnetworks, stage supervision and self-distillation can only improve the performance of the main network, while the proposed stimulative training can achieve the highest performance of main network and the highest average performance of all subnetworks. (2) As shown in Fig. r8 (a), (b), (c) and (d) of the revised supplementary, the proposed stimulative training can better relieve the network loafing problem than all the other methods. (3) As shown in Fig. r4, r5, r6 and Fig. r7 of the revised supplementary, the proposed stimulative training can provide stronger robustness in resisting various network destruction operations than all the other methods.
>
> Besides these experimental results, we find that: 1) The improved performance of Stochastic Depth can be also interpreted as relieving the loafing problem defined in this work; 2) the proposed stimulative training is actually complementary to layer/stage supervision and Self-Distillation, and their combinations can be a worthy research direction in the future.
>
> [7] Zhang L, Song J, Gao A, et al. Be Your Own Teacher: Improve the Performance of Convolutional Neural Networks via Self Distillation[J]. arXiv preprint arXiv: 1905.08094, 2019.
>
> [8] Huang G, Sun Y, Liu Z, et al. Deep networks with stochastic depth[C]//European conference on computer vision. Springer, Cham, 2016: 646-661.
>
> **Table r1: Comparisons**
> |Method|Time|Memory|Main(%)|All(%)|
> |:--------------|:----------- |:------ |:-----------|:----|
> | CT|	16.91h|	3291MiB	|77.39	|55.26±13.37|
> | CT + layer supervision |	23.3h|	7193MiB	|78.77|	59.18±11.12|
> | CT + stage supervision |	19.3h|	5197MiB	|78.59|	54.82±13.31|
> | Self-Distillation	|26.8h|	3887MiB	|79.59	|50.39±14.22 |
> |Stochastic Depth|	13.6h|	3291MiB	|78.43|	70.72±3.76|
> |ST	|24.08h	|3291MiB|	81.07|	80.01±0.59|
>
> ***Q4:limitation section***
>
> A4: Thanks for the helpful suggestion, we have added the limitations section in the revised supplementary to discuss possible issues, applications and societal impact of the proposed method.
>
> The proposed method suffers from about 1.4 times of computation cost of the original model training to get better performance and robustness. As the first research of network loafing problem, the proposed method is a positive pioneer-like exploration. We believe designing a more efficient method to solve the network loafing problem is a worthy research direction in the future.
>
> Residual structure is widely applied in numerous different types of models including DenseNet and transformer. It will be of vital value to study whether the loafing problem exists in these models and explore the proper method to solve this problem. Since most of existing works overlook the loafing problem, it may also be a feasible way to apply the proposed method in pretrained model. What’s more, the proposed method adopts main network logits as supervision to alleviate loafing and analogously, the proposed method may be applied in self-supervised learning.
>
> We believe taking full advantage of splendid achievements in interdisciplinary research can help promote the development of deep learning. We hope this paper can provide a new perspective to inspire more researchers to comprehend and improve DNNs from other fields such as social psychology.

---

> ### Author Response · Authors · 2022-08-02
> **Response to Reviewer 6eTg**
>
> Thanks for your review comments.
>
> ***Q1: Framing. Is the grounding of the work in social psychology fundamentally important to the presented approach?***
>
> A1: (1) In the manuscript, we introduce the concept of social loafing with “individual/member in a social group” instead of “people in a social group”, to avoid possible ethical issues. In fact, social loafing is a widespread social psychology phenomenon, which has been verified in kinds of social groups (people as well as animals) [1-3].
>
> (2) In this paper, we find that the loafing phenomenon also exists in DNNs area, for the first time. The brand-new perspective can help us to further understand various deep models (e.g., ResNet and DenseNet) and learning methods (e.g, OFA NAS and Stochastic Depth), which in turn can inspire new directions and works.
>
> (3) Celebrated Dropout [4] is also motivated by a theory of the role of sex in evolution and utilized to reduce the co-adaptation problem, our paper is analogous. We have no intention to contribute to false AI hype and other ethical issues but hope to reveal the generality of different areas.
>
> (4) The social psychology perspective and the loafing problem are considered creative (R1), interesting (R1, R2, R4) and novel (R1, R4) by other reviewers, we hope R3 can consider them.
>
> [1] Ingham A G, Levinger G, Graves J, et al. The Ringelmann effect: Studies of group size and group performance[J]. Journal of experimental social psychology, 1974, 10(4): 371-384.
>
> [2] Simms A, Nichols T. Social loafing: A review of the literature[J]. Journal of Management, 2014, 15(1): 58-67.
>
> [3] Phonekeo S, Dave T, Kern M, et al. Ant aggregations self-heal to compensate for the Ringelmann effect[J]. Soft Matter, 2016, 12(18): 4214-4220.
>
> [4] Srivastava, Nitish, et al. "Dropout: a simple way to prevent neural networks from overfitting." The journal of machine learning research 15.1 (2014): 1929-1958
>
>
> ***Q2: Learning redundant blocks. How can the objective of learning redundant residual blocks be combined with the understanding that CNNs extract feature hierarchies?***
>
> A2:(1) It is widely accepted that modern CNNs such as MobileNet and ResNet have multi-stage structure where sequential stages (usually different resolutions between stages and the same resolution in one stage) generate hierarchical representations. From this view, whatever the proposed sampling method, deleting and shuffling strategy, they don’t break the multi-stage structure, which means that the hierarchical structure of representations is reserved.
>
> (2) Moreover, [5] and [6] discover that successive layers in the same stage of residual networks are in fact estimating the same optimal feature map so that the outputs of these layers stay relatively close to each other at convergence. From this view, the proposed method actually forces each layer in the same stage of residual networks to independently estimate the optimal feature map, so as to learn better feature representation in each stage. A more proper comprehension of the proposed method is to train better single blocks instead of training redundant blocks.
>
> (3) In fact, what we attempt to emphasize in the analysis is that our method not only can improve the performance but also the robustness in resisting various network destruction operations. In the practical application, we believe it’s important to ensure the running network won’t collapse due to some layers’ damage.
>
> (4) The key point of the proposed method is to provide each member (i.e., subnetworks) with appropriate supervision and make them has the consistent overall goal with the group (i.e., main network), instead of forcing each member to do the same job.
>
> (5) Table 1 in the main text show that the proposed method can maintain excellent performance and robustness on different models (e.g., MobileNet and ResNet) and datasets (e.g., CIFAR10, CIFAR100 and ImageNet-1K).
>
> [5] Greff K, Srivastava R K, Schmidhuber J. Highway and residual networks learn unrolled iterative estimation[J]. arXiv preprint arXiv:1612.07771, 2016.
>
> [6] Veit A, Wilber M J, Belongie S. Residual networks behave like ensembles of relatively shallow networks[J]. Advances in neural information processing systems, 2016, 29.

---

### Official Review · Reviewer_g3Zh · 2022-07-11

**Rating:** 4
**Confidence:** 2
**Soundness:** 2 fair
**Presentation:** 2 fair
**Contribution:** 3 good

**Summary:**

This paper explores the use of a training method for residual networks called "stimulative training". This method is inspired by the psychological phenomenon of "social loafing" where members of a group give less effort when in a group compared to when not in a group. The paper hypothesizes a similar "network loafing" phenomenon for neural networks where subnetworks in a larger network would perform suboptimally compared to when a similar network is trained by itself. The paper devises a training strategy called stimulative training that samples random subnetworks and minimizes a kl divergence loss between the output of the sampled subnetwork and the full network. The paper performs a few experiments to show that
- The loafing problem exists with traditional training strategies and is eliminated/reduced with their method
- Their method improves the robustness of the network to deletion/permutation of layers
- Improves the overall performance of the network for CIFAR 100 and Imagenet datasets.

**Questions:**

- When you mention sampling x subnetworks (eg sampling 144 subnetworks), is that on each step? If so, what is the difference in training time between the two methods?
- In section 3.3, when discussing the ordered residual sampling, the justification given for why ordered residual sampling is better than raw sampling is that because of the size of the space, raw sampling isn't able to train all the subnetworks enough. Does that mean that if you trained the method with raw sampling for longer, performance would continue improving until it beat the ordered residual sampling? Is the learning curve for raw sampling converged, or is it still improving?
- Is the sampling method uniform sampling?
- Did you try experiments with MSE instead of KL divergence for the stimulative training? Was there a difference?

**Limitations:**

Other than showing reproducibility, I'd mostly want to see a comparison to/discussion of a method like self-distillation, which is not limited by the type of architecture, which this method is. Assuming the network loafing phenomenon exists with other architectures as well, why shouldn't we use something like self-distillation instead of this method, since it's more general? Is there a big improvement in the ease of use or performance with resnets when using this method? Also, I am curious as to how much longer this method takes compared to conventional training.

**Strengths And Weaknesses:**

Strengths:
- Connecting the training of neural networks to the social loafing phenomenon was an interesting point. Showing that simply getting each individual subnetwork more supervision to a common goal leads to an improvement in performance is also an interesting result, as I could imagine that it could lead to worse generalization on test set (something like the group think phenomenon).
- I think the types of experiments selected was good. They covered a range of questions about this phenomenon.

Weaknesses:
- As far as I can tell, the results are based on only a single run of data. To draw the conclusions that the paper wants us to, I think they need to show that the results are replicable. For example, even if we say network loafing exists, it's unclear if we can conclude that it is a problem with a single run of data. Table 1 shows the results comparing the final accuracy of conventional training with stimulative training, and the improvement is small. To conclude that it is significant, the paper needs to show that it is replicable.
- The method seems fairly similar to self distillation. In self distillation, they have some added machinery to be able to use the method on architectures where the output of each layer might not be the same size. In this paper, because the architectures are limited to residual networks, that machinery is not needed. Still, it seems to be a special case of self distillation for residual networks (although the paper does examine the specific phenomenon of network loafing). It would be interesting to see something like self-distillation as a baseline.
- For Figure 4, please use more distinctive markers. It's difficult to tell the difference between markers of the same color.

---

> ### Author Response · Authors · 2022-08-02
> **Response to Reviewer g3Zh**
>
> ***Q4: what is the difference in training time between the two methods***
>
> A4: More specifically, we only randomly sample one subnetwork on each step. Following the reviewer’s suggestion, we further show the difference in training time and memory consumption between the two methods, as shown in Table r5. For training time, ST is about 1.4 times (but less than 2 times) that of CT, since ST employs the main network and a sampled subnetwork at each step and the sampled subnetwork usually takes much less time than the main network. For memory consumption, ST and CT are basically the same, since each subnetwork is sampled from the main network.
>
> **Table r5: Training time and memory consumption**
> |Method| MBV3_C10| MBV3_C100 |Res50_C100|
> |:--------------|:----------- |:------ |:-----|
> | CT (time)|16.77h	|16.91h	|15.28h|
> | ST (time)	|23.64h	|24.08h	|21.52h|
> |CT (memory)|	3361MiB	|3291MiB|	4647MiB|
> |ST (memory)|	3361MiB	|3291MiB|	4647MiB|
>
> ***Q5: the justification given for why ordered residual sampling is better than raw sampling is that because of the size of the space, ...***
>
> A5: (1) The main reason that Raw Sampling (75.34%) performs much lower than Stochastic Residual Sampling (80.53%) and Ordered Residual Sampling (81.07%) is that, it introduces a mass of single branch sub-networks with different depths. These single branch networks are hard to optimize, and the deeper the network, the harder it is to converge, due to the loss in information flow [1][2]. For example, the deepest vgg [3] is only 19 layers, and its performance is much lower than resnet18 [1]. Thus, training a model with raw sampling for longer time may slightly improve its performance, but it wouldn’t beat the ordered residual sampling under limited computation resources
>
> (2) The main reason that Ordered Residual Sampling (81.07%) performs better than Stochastic Residual Sampling (80.53%) is that, the former can noticeably reduce the size of sampling space (from O(2^n) to O(n)), making it possible to train each sub-network sufficiently under limited computation resources. In addition, from the perspective of neural differential equations [4], ordered residual sampling will maintain the continuity of residual networks.
>
> [1] He, K., Zhang, X., Ren, S., Sun, J.: Deep residual learning for image recognition. arXiv preprint arXiv:1512.03385, 2015.
>
> [2] Srivastava, R.K., Greff, K., Schmidhuber, J.: Highway networks. arXiv preprint arXiv:1505.00387, 2015.
>
> [3] Simonyan K, Zisserman A. Very deep convolutional networks for large-scale image recognition[J]. arXiv preprint arXiv:1409.1556, 2014.
>
> [4] Chen R T Q, Rubanova Y, Bettencourt J, et al. Neural ordinary differential equations[J]. Advances in neural information processing systems, 2018, 31.
>
>
> ***Q6: Is the sampling method uniform sampling?***
>
> A6: More specifically, we randomly sample from the sampling space, and all subnetworks in the sampling space have the same sampling probability.
>
> ***Q7: Did you try experiments with MSE instead of KL divergence for the stimulative training?***
>
> A7: Following the reviewer’s suggestion, we further try experiments with MSE instead of KL divergence for the stimulative training. As shown in Table r6, using MSE for the stimulative training can still improve the performance of common training, while using KL divergence for the stimulative training can achieve the best performance on various models and datasets.
>
> **Table r6: Comparisons**
> |Method| MBV3_C10| MBV3_C100 |Res50_C100|
> |:--------------|:----------- |:------ |:-----|
> | CT|	95.72	|77.39	|76.53|
> | ST(MSE)	|96.47	|78.78|	78.12|
> |ST(KL)	|96.88|	81.07|	81.06|

---

> ### Author Response · Authors · 2022-08-02
> **Response to Reviewer g3Zh**
>
> Thanks for your review comments.
>
> ***Q1: To conclude that it is significant, the paper needs to show that it is replicable.***
>
> A1: (1) To show that the results are replicable, we compare the main network performance of stimulative training (ST) with that of common training (CT) on various residual networks and datasets, and each setting is run for five times. The average results are shown in Table r1. As we can see, ST can robustly achieve much higher performance than CT, on various residual networks and datasets.
>
> (2) To show that the loafing problem are replicable, we further verify that DenseNet also suffers from the loafing problem. Table r2 and Table r3 show the results of different DenseNet networks which are trained on ImageNet and CIFAR100 respectively. As we can see that, different DenseNet networks invariably suffer from the loafing problem, that is, the sub-networks working in a given DenseNet network are prone to exert degraded performances than these sub-networks working individually. Moreover, the loafing problem of deeper DenseNet networks is inclined to be more severe than that of shallower ones, that is, the same sub-network in deeper DenseNet networks constantly presents inferior performance than that in shallower DenseNet networks. As Fig. 1 of the main text has shown that different ResNet networks invariably suffer from the loafing problem, we can conclude that various residual networks invariably suffer from the loafing problem on various datasets.
>
> **Table r1: The main network performance (%) after ST CT. Each setting is run for five times.**
> |Method| MBV3_C10| MBV3_C100 |Res50_C100|
> |:--------------|:----------- |:------ |:-----|
> | CT | 95.72±0.35|	77.32±0.19|	76.80±0.26|
> | ST | 96.83±0.037|	81.11±0.25|	81.03±0.13|
>
> **Table r2:DenseNet networks trained on ImageNet**
> |Main-net\Sub-net|	DenseNet121	|DenseNet169|	DenseNet201|
> |:------ |:-----------|:---|:---|
> |DenseNet121|	74.86|	20.91|	11.57|
> |DenseNet169	|-|	76.46|	51.18|
> |DenseNet201	|-	|-|	77.44|
>
> **Table r3:DenseNet networks trained on CIFAR100**
> |Main-net\Sub-net|	DenseNet121	|DenseNet169|	DenseNet201|	DenseNet264|
> |:------ |:-----------|:---|:---|:---|
> |DenseNet121|	78.84|	43.64|	31.51|	10.01|
> |DenseNet169|	-|	79.64|	70.78|	48.41|
> |DenseNet201|	-|	-|	79.77|	62.29|
> |DenseNet264|	-|	-|	-|	79.81|
>
>
> ***Q2: The method seems fairly similar to self distillation. It would be interesting to see something like self-distillation as a baseline.***
>
> A2: In this paper, we first demonstrate the loafing problem of residual networks and then propose the stimulative training to relieve this problem. Although similar, there exits several key differences between stimulative training and self-distillation: Technically, self-distillation methods usually need to introduce additional networks or structures, and employ fixed teacher-student pairs. As a comparison, our method does not require any additional network or structure, and the student network is a randomly sampled sub-network of a network. On target, our method is essentially designed to address the loafing problem of residual networks, which have not been addressed in the literature so far as we know. In addition, the proposed stimulative training is actually complementary to self-distillation and can be seamless combined. Experimentally, we further compare the proposed stimulative training with the classical self-distillation [1]: (1) The comprehensive comparisons are shown in Table r4. As we can see, self-distillation can only improve the performance of the main network, while the proposed stimulative training can improve both the performance of the main network and the average performance of all subnetworks. (2) As shown in Fig. r8 (a) of the revised supplementary, the proposed stimulative training can better relieve the network loafing problem than self-distillation. (3) As shown in Fig. r4 of the revised supplementary, the proposed stimulative training can provide stronger robustness in resisting various network destruction operations than self-distillation.
>
> [1] Zhang L, Song J, Gao A, et al. Be Your Own Teacher: Improve the Performance of Convolutional Neural Networks via Self Distillation[J]. arXiv preprint arXiv: 1905.08094, 2019.
>
> **Table r4:Comparisons**
> |Method|Time|Memory|Main(%)|All(%)|
> |:--------------|:----------- |:------ |:-----------|:----|
> | CT|	16.91h|	3291MiB	|77.39|	55.26±13.37|
> | Self-Distillation [1]	|26.80h	|3887MiB|	79.59|	50.39±14.22|
> | ST|	24.08h	|3291MiB|	81.07|	80.01±0.59 |
>
> ***Q3: For Figure 4, please use more distinctive markers.***
>
> A3: Thanks for the helpful suggestion, we will use more distinctive markers in Fig. 4 of the revised manuscript.

---

### Official Review · Reviewer_yqSA · 2022-07-14

**Rating:** 6
**Confidence:** 3
**Soundness:** 3 good
**Presentation:** 3 good
**Contribution:** 3 good

**Summary:**

This paper finds an interesting problem with the residual networks, i.e., the network loafing problem. They find that the sub-networks will perform worse than those that are individually trained. Inspired by social psychology, they propose a stimulative training strategy, which randomly samples a residual sub-network and calculates the KL-divergence loss between the sampled sub-network and the residual network, to provide additional supervision to the sub-network. The proposed strategy is validated by a comprehensive theoretical and empirical analysis.

**Questions:**

1. Can you explain why different sampling schemes make such a big difference (~5% in accuracy, figure 3)? It seems that in order for the main network to perform well, the subnetworks need to contain residual links inside. It will be great to see the explanation to this or see a comparison with normal training vs random sampling.
2. How do you measure the performance of subnetworks? Will you retrain the last logits layer? If you do not retain the last logits layer, then the performance drop is understandable because the range/distribution of the input data to the last logits layer has changed. Even if the last logits layer is not retrained, I would expect some scaling/correction when the residual connection is removed (removing the residual connection will somewhat downscale the signal). I think the performance of a subnetwork should be evaluated in a fair way and we need to include the necessary corrections. Because what we would like to know is how well the subnetwork itself is to perform classification and we should minimize the impact from other factors (e.g., the scaling effects).
3. I am curious about how does the proposed stimulative training strategy compared with providing supervision(the class label) directly to each layer? I am wondering if the authors directly compute CE loss between the labels and intermediate output instead of the KL loss, how the results would be like? I would like to know if the supervision to intermediate layers are the key to improve the performance and how effective the KL loss is.


**Limitations:**

See the questions above.

**Strengths And Weaknesses:**

Originality: I like the idea to view the residual network from a social psychological perspective. This perspective is creative. Following the "unraveled view", they see the residual network as an ensemble of sub-networks and study the performance of each sub-network. They discover a novel problem called "network loafing", which has analogous to the problem in social psychology where an individual contributes less when in a group.
Quality:
In general, the paper is technically sound.
Clarity: Overall, the paper is well written and easy to understand. The motivation and the contributions are clear, and the whole paper is organized well.
Some minor issues with the writing: Some experimental details are missing, e.g., how do you measure the performance of subnetworks? Will you retrain the last logits layer?

---

> ### Author Response · Authors · 2022-08-02
> **Response to Reviewer yqSA**
>
> ***Q3: How does the proposed stimulative training strategy compared with providing supervision (the class label) directly to each layer?...***
>
> A3: Following the reviewer’s suggestion, we further compare the proposed stimulative training with providing supervision (the class label) directly to each layer or each stage: (1) The comprehensive comparisons are shown in Table r2. As we can see, layer supervision can improve both the performance of the main network and the average performance of all subnetworks, stage supervision can only improve the performance of the main network, while the proposed stimulative training can achieve the best performance of main network and the best average performance of all subnetworks. (2) As shown in Fig. r8 (c) and (d) of the revised supplementary, the proposed stimulative training can better relieve the network loafing problem than layer supervision and stage supervision. (3) As shown in Fig. r6 and Fig. r7 of the revised supplementary, the proposed stimulative training can provide stronger robustness in resisting various network destruction operations than layer supervision and stage supervision. Moreover, the proposed stimulative training is actually complementary to layer supervision and stage supervision and can be seamlessly combined.
>
> **Table r2: Comparisons.**
> |Method|Time|Memory|Main(%)|All(%)|
> |:--------------|:----------- |:------ |:-----------|:----|
> | CT | 16.91h | 3291MiB| 77.39|55.26±13.37|
> | CT + layer supervision | 23.3h |7193MiB |78.77 |59.18±11.12|
> | CT + stage supervision | 19.3h | 5197MiB |78.59 |54.82±13.31|
> | ST | 24.08h | 3291MiB | 81.07 | 80.01±0.59 |

---

> ### Author Response · Authors · 2022-08-02
> **Response to Reviewer yqSA**
>
> Thanks for your review comments.
>
> ***Q1: why different sampling schemes make such a big difference?...***
>
> A1: The main reason that Raw Sampling performs much lower (about 5% in accuracy, Fig. 3) than Stochastic Residual Sampling and Ordered Residual Sampling is that, it introduces a mass of single branch sub-networks with different depths. These single branch networks are hard to optimize, and the deeper the network, the harder it is to converge, due to the loss in information flow [1][2]. For example, the deepest vgg [3] is only 19 layers, and its performance is much lower than resnet18[1]. In practice, sampling residual networks (e.g., Stochastic Residual Sampling and Ordered Residual Sampling) are better choices to facilitate the training of sub-networks with different depths and improve the final performance. Therefore, MBV3 with Raw Sampling only has a Top 1 accuracy of 75.34% (shown in Fig 3). As a comparison, MBV3 with common training has a Top 1 accuracy of 77.39% (shown in Table 1), while MBV3 with Stochastic Residual Sampling or Ordered Residual Sampling has a Top 1 accuracy of 80.53% or 81.07% (shown in Fig 3).
>
> [1] He, K., Zhang, X., Ren, S., Sun, J.: Deep residual learning for image recognition. arXiv preprint arXiv:1512.03385, 2015.
>
> [2] Srivastava, R.K., Greff, K., Schmidhuber, J.: Highway networks. arXiv preprint arXiv:1505.00387, 2015.
>
> [3] Simonyan K, Zisserman A. Very deep convolutional networks for large-scale image recognition[J]. arXiv preprint arXiv:1409.1556, 2014.
>
>
> ***Q2: How do you measure the performance of subnetworks? ...***
>
> A2: When measuring the performance of subnetworks, we will employ batch-norm re-calibration for each sampled sub-network, following [4]. (As shown in Appendix C.2 of supplementary). Note that both common and stimulative training keep the same testing settings. Since batch-norm re-calibration can be considered as the necessary correction to ensure a fair evaluation, we don’t retrain the last logits layer or upscale the signal in the original paper. In this response, we also provide some experimental results after retraining the last logits (with 10/100 epochs) and upscaling the signal (following [5]), as shown in Table r1. On the one hand, retraining the last logits can further improve the subnetworks after CT, but subnetworks from (ST + retrain the last logits) still perform much better than that from (CT + retrain the last logits). Besides, we show that retraining the last logits with different epochs has no effect on the relative ranking of subnetworks, and the performance of retraining 10 epochs is similar to that of retraining 100 epochs. On the other hand, we can see that upscaling the signal has little impact on the performance, the main reason is that batch-norm re-calibration can correct the signal automatically.
>
> [4] Jiahui Yu and Thomas S Huang. Universally slimmable networks and improved training techniques. In Proceedings of the IEEE/CVF international conference on computer vision, pages 1803–1811, 2019.
>
> [5] Srivastava, N., Hinton, G., Krizhevsky, A., Sutskever, I., Salakhutdinov, R.: Dropout: A simple way to prevent neural networks from overfitting. The Journal of Machine Learning Research 15(1) (2014) 1929–1958
>
> **Table r1: Top 1 Accuracy (%) after retraining or upscaling**
> ||Subnet1  | Subnet2      | Subnet3     |
> |:--------------|:----------- |:------ |:-----------|
> |**Method**|[1,1,1,1,1]|[2,2,2,1,1]|[2,3,3,2,2]|
> ||(96.69M)|(129.07M)|(192.65M)|
> | CT | **28.48** | **36.13**| **65.25**|
> | CT + retrain the last logits (10) |  61.44 |66.87 |74.95 |
> | CT + retrain the last logits (100)| 63.54| 66.7 |75.2 |
> | CT + upscale the signal | 28.48 | 36.9 |65.16 |
> | ST | **77.85** | **79.43** | **80.61** |
> | ST + retrain the last logits (10) | 77.97 | 79.28 | 80.52|
> | ST + retrain the last logits (100) | 78 | 79.26 | 80.7 |
> | ST + upscale the signal | 77.85 | 79.12 | 80.06 |

---

### Review · Ethics_Reviewer_98dH · 2022-08-05

**Recommendation:**

I'd recommend that the authors include a statement showing that they have reflected on whether there is any potential negative social impact to their work (think about potential applications where it might be used). It sounds like they have already done this per the author response comments.

**Ethical Issues:**

Yes

**Ethics Review:**

This paper was flagged because it did not discuss the potential negative societal impact or limitations as outlined in the Ethics Guideline.

---

### Review · Ethics_Reviewer_HFgQ · 2022-08-05

**Recommendation:**

The authors could edit instances of wording that equate network behavior to the behavior of humans or animals in order to clarify the metaphorical relationship between the work at hand and the social psychology literature. They could also provide an explicit statement that because the connection to social psychology is metaphorical, the paper's experimental results do not provide any insight back into or implications for social loafing as defined for people and animals. If possible, the limitations section should be moved to the main body of the paper where it will be most beneficial to the audience. Additionally, I wonder if the proposed training method has implications for how easily training data instances might be recovered from trained models, which may lead to privacy concerns with image datasets that contain photos of people, like ImageNet.

**Ethical Issues:**

Yes

**Ethics Review:**

The authors present a new characterization of residual networks using an analogy from social psychology, and propose a novel training method following this inspiration. The concern is that this analogy anthropomorphizes the networks and contributes to AI hype. Additionally, the authors do not include a limitations section in the main body of their paper.

---

### Meta-Review · Area_Chair_GYeQ · 2022-08-26

**Recommendation:** Accept
**Confidence:** Less certain

**Metareview:**

This paper proposes to study the "loafing" problem in deep ResNets, which suggests that the sub networks of a deep ResNet perform significantly worse than the same architecture trained alone. It proposes a simple technique which jointly trains the main network and the KL divergence the main network's output and that of a random subnetwork. It is shown empirically that this technique improves the final accuracy for both the main network and random subnetworks. The reviewers agreed that the "loafing" problem is an interesting phenomenon, but also raised concerns about both the motivation/presentation and comparison with similar techniques like deep supervised and self distillation. The authors provided extensive responses with new additional experimental results. After the discussion phase, the reviewers reached to a consensus of acceptance, conditioned on that the authors carefully address the framing of "loafing" and make clear that the "loafing" term is just a loose analogy without any real implications to biology. The AC agrees that the problem identified in this paper is interesting and can have implications to both regularization and model compression. However the authors should try to remove the excessive reference to the social psychology aspects, which does provide scientific justification to the method but rather could invite unnecessary confusion and controversy.

**Award:**

No

---

### Decision · Program_Chairs · 2022-09-14

Accept